# Visualizing the gas-sensitive structure of the CuZn surface in methanol synthesis catalysis

Sigmund Jensen[1,3], Mathias H. R. Mammen[1,3], Martin Hedevang[1], Zheshen Li [2], Lutz Lammich [1,2] & Jeppe V. Lauritsen [1] ✉

Methanol formation over Cu/ZnO catalysts is linked with a catalytically active phase created by contact between Cu nanoparticles and Zn species whose chemical and structural state depends on reaction conditions. Herein, we use variable-temperature scanning tunneling microscopy at elevated pressure conditions combined with X-ray photoelectron spectroscopy measurements to investigate the surface structures and chemical states that evolve when a CuZn/Cu(111) surface alloy is exposed to reaction gas mixtures. In $CO_2$ hydrogenation conditions, Zn stays embedded in the CuZn surface, but once CO gas is added to the mixture, the Zn segregates onto the Cu surface. The Zn segregation is CO-induced, and establishes a new dynamic state of the catalyst surface where Zn is continually exchanged at the Cu surface. Candidates for the migrating few-atom Zn clusters are further identified in time-resolved imaging series. The findings point to a significant role of CO affecting the distribution of Zn in the multiphasic ZnO/CuZn/Cu catalysts.

Catalytic methanol production ($CH_3OH$) receives significant attention due to the prospects of converting captured $CO_2$ and green $H_2$ from electrolysis into a liquid and portable renewable fuel[1,2]. Heterogenous catalysts based on a mixture of Cu, ZnO and $Al_2O_3$ (CZA) are employed at the industrial scale for methanol production from synthesis gas ($H_2$, $CO_2$, and CO)[2,3]. For the CZA catalyst, the intrinsic catalytic activity is associated with the Cu and ZnO, while the $Al_2O_3$ component is considered a structural stabilizer[3]. The Cu/ZnO system represents an intriguing and intensively studied example of catalytic synergy between a metal (Cu) and a metal oxide (ZnO), since the catalytic turnover rate towards methanol is strongly promoted when these two compounds are brought into contact[4,5]. Despite intense research, the role of Zn in the catalytic mechanism and the nature of the active site has remained elusive. It has only recently been possible to gain consistent information on the active chemical state of the catalyst thanks to in-situ catalyst characterization methods[6–8]. From the as-synthesized state of the catalyst, consisting of separate Cu and ZnO particles, it is concluded that the introduction of reductive reaction conditions (150–200 °C and $CO_2$/CO and $H_2$ at 50–100 bar) initiates spill-over of Zn species onto the metallic Cu, thus activating the catalytic synergy[9–11].

Methanol formation is possible through both hydrogenation of the CO and $CO_2$, but isotope labeling experiments have indicated that the dominant reaction pathway to methanol is from the $CO_2$[12–14]. Methanol synthesis based entirely on point-source $CO_2$ gas as the carbon source would in fact be highly desirable, but the addition of CO to the $CO_2$/$H_2$ gas feed is, however, a requirement for stable high-turnover conditions for the CZA catalyst. The role of CO in the gas feed appears to be complex showing evidence of both a promoting[15,16] and inhibiting role[17,18] depending on reaction conditions[19]. The positive effect of CO has been attributed to its role first in the reduction of the ZnO component to activate the Cu-Zn synergy[20,21] and as a scavenger of strongly inhibiting oxygen and water species (a product from $CO_2$ hydrogenation) through the water-gas shift reaction (WGS)[22,23]. The inhibiting effect of CO is not understood but is often speculated to be caused by competitive adsorption of CO in the $CO_2$ hydrogenation pathway at the active site[24].

These observations strongly indicate that the chemical and structural state of the catalytic interface is highly sensitive to the gas composition[25]. The chemical state of the catalyst's surface has been proposed to reflect either an interfacial Zn oxide film on Cu formed

[1]Interdisciplinary Nanoscience Center (iNANO), Aarhus University, 8000 Aarhus C, Denmark. [2]Department of Physics and Astronomy, Aarhus University, 8000 Aarhus C, Denmark. [3]These authors contributed equally: Sigmund Jensen, Mathias H. R. Mammen. ✉e-mail: jvang@inano.au.dk

by migration of reduced ZnO[26–28] or a CuZn alloy surface resulting from ZnO reduction and dissolution of Zn into Cu[5,24,29]. Transmission electron microscopy (TEM) observations in mbar pressure conditions have linked dynamic transitions at the Cu/ZnO interface directly to variations in methanol formation activities[25,30,31]. Moreover, a surface sensitive in-situ method based on ambient-pressure X-ray photoelectron spectroscopy (AP-XPS) was recently applied by Amann et al.[8] to demonstrate that both oxide and alloy phases seem to co-exist on a Zn/ZnO/Cu(211) model catalyst. Furthermore, formation of methoxy and formate species could be observed on this type of model system, in agreement with IR spectroscopy results[32,33]. It was further shown that the presence of CO promotes the formation of a CuZn alloy phase over ZnO/Cu which dominates in pure $CO_2$ hydrogenation conditions ($H_2/CO$ vs. $H_2/CO_2$)[8]. This and other recent work support a CuZn phase[34] as a candidate to describe the active phase, in line with the original work of Nakamura et al. [4]. The lack of information on the actual surface structures generated under such conditions, however, prevents direct assessment of the substantial number of different models for the actual active site for methanol formation that have been proposed[6,7,26,28,29]. In the context of understanding the active state at the atomic scale and its sensitivity towards reacting, the aim of this work is to provide in-situ and atomic-scale surface imaging of the proposed active CuZn phase and its structural evolution in varying reactant atmospheres.

Scanning tunneling microscopy (STM) has been applied to planar catalyst model systems to elucidate many atomic-scale aspects of surface structure and dynamics of surface reactions[35,36], including work on CuZn alloys[37–40], ZnO/Cu[27,41–43] and Cu/ZnO[44–47]. STM studies at elevated pressure conditions are referred to as near-ambient pressure STM (NAP-STM) (or ambient pressure/high-pressure STM) and offer the possibility to observe the structure of planar model systems in pressurized conditions that bridge the gap towards catalytic conditions[48–51]. The STM technique is not fundamentally limited by the pressure, but for the practical application the accessible pressure range in NAP-STM has so far typically been from a few mbar to bar pressure, which is lower than the 50–100 bar applied in industrial methanol catalysis. Wang et al. [52] used NAP-STM at room temperature to monitor how monolayer ZnO films on Cu(111) had undergone large morphological changes after being exposed to 10 mbar $CO_2$ hydrogenation conditions.

Herein, we prepared a well-defined CuZn alloy phase on a Cu(111) surface and use variable-temperature NAP-STM experiments to image how Zn species in this system respond in-situ while it is brought up in temperature and gas pressure (3–10 mbar) towards methanol synthesis reaction conditions. Our experiments reveal a profound effect of the gas composition on the surface composition and morphology. In $CO_2/H_2$ gas, our atom-resolved NAP-STM images show that the Zn stays stable and embedded in the Cu surface as a CuZn alloy, implying that the CuZn phase is stable in pure $CO_2$ hydrogenation conditions. In contrast, when the CO is added to the $H_2/CO_2$ gas feed (corresponding to syngas), we observe that Zn segregation becomes strongly activated. The NAP-STM imaging reveals a surface morphology where Zn coexists in two distinct phases consisting of a reduced monolayer Zn phase with a fractional O coverage ($Zn-O_x$) and a CuZn alloy near the perimeter of the Zn monolayer islands. Our NAP-STM results point in detail to a highly dynamic catalytic interface in methanol synthesis conditions, where Zn species are shuttled between the alloy and surface Zn. These processes are controlled by a balance between the favorable Zn alloying into Cu and a CO-driven extraction and segregation of Zn from the CuZn alloy.

## Results
### CuZn alloy on Cu(111)
Our experiments are based on a planar model system consisting of a CuZn surface alloy synthesized on the surface of a clean Cu(111) single

crystal (Fig. 1A). The pristine CuZn/Cu(111) surface used as the starting point in all our experiment is prepared under ultra-high vacuum (UHV) by first depositing $0.15 \pm 0.03$ monolayer (ML) of Zn, corresponding to the relevant amount of Zn on Cu reported by ref. 5. The Zn deposition was done in a separate deposition chamber connected to the STM microscope (see Methods)[27,53]. The uniform CuZn alloy surface is formed by a subsequent annealing to 500 K, resulting in the surface structure in Fig. 1A. The contrast in atomically resolved STM images involves an electronic contribution originating from the local density of states (LDOS) that often makes it possible to discriminate the individual atomic species in alloys[54,55]. In accordance, our atomically resolved UHV-STM image of the resulting CuZn/Cu(111) surface reveals a Cu(111) lattice where isolated bright protrusions are located on the Cu positions, reflecting that the surface now consists of randomly distributed Zn atoms substitutionally alloyed into the Cu(111) matrix[37,56]. In agreement, in Fig. 1B the Zn LMM Auger peak recorded at a synchrotron light source shows the presence of metallic $Zn^0$ in the surface region.

For direct Zn deposition at room temperature without the annealing step, the Zn adatoms nucleate near the Cu(111) step edges as coherent adlayer Zn islands[57] (termed $Zn_{ad}$) (Fig. 1C and Supplementary Fig. 1). The monolayer $Zn_{ad}$ islands are discerned by a Zn step height of $h_{Zn}$ - 1.9 Å, slightly lower than that of the Cu step at $h_{Cu}$ - 2.1 Å (see STM line profiles in Supplementary Fig. 2 and direct comparison in Supplementary Fig. 1). A previous study in ref. 57 has shown that intermixing of Zn into the topmost layer of the Cu(111) surface occurs exclusively at the $Zn_{ad}$ island perimeter, leading to a sharp boundary between alloy and Cu, seen from the distinct darker STM contrast of the mixed CuZn zone highlighted in Fig. 1C compared with the bare Cu(111). The structural and spectroscopic features of these Zn islands and the intermixed CuZn zone around the Zn island in Fig. 1C will become important for our assignment of the Zn state observed in-situ in the NAP-STM experiments in the following.

We also monitored the $Zn3d$ valence band spectrum which is shown (Fig. 1D) to be responsive to the formation of the alloy phase, leading to a peak shape change during a temperature series. The transition from $Zn_{ad}$ islands to the CuZn alloy is seen by a narrowing and distinct sharpening of the spin-split Zn $3d$ states as the annealing temperature is increased from RT towards 500 K (Fig. 1D). This change in peak structure can be explained by the gradual elimination of hybridized Zn–Zn bonds in the monolayer $Zn_{ad}$ islands as the alloy is formed, thus converting a widened Zn $d$-band (blue spectrum) into isolated Zn atoms in the Cu with a sharp Zn $3d$ signature (black spectrum)[8,58,59].

### CuZn in $CO_2$ hydrogenation conditions
Next, we investigate the structure of the CuZn/Cu(111) surface in $CO_2$ hydrogenation conditions by NAP-STM. The experiment is conducted by first imaging the CuZn alloy sample in UHV, followed by the introduction of a pre-mixed $H_2/CO_2$ (2:1) gas at 3 mbar and NAP-STM imaging at selected temperatures from 300 K to 423 K. At the end of the NAP-STM experiments, the sample is cooled and the NAP-STM cell is evacuated back to vacuum conditions whereafter the sample is characterized in UHV. The introduction of the $H_2/CO_2$ gas mixture at room temperature is not seen to induce any pronounced morphological change to the surface compared with the pristine CuZn surface. The intact alloy state of the CuZn surface is verified by atomically resolved in-situ STM image in Fig. 2C (10 mbar $H_2/CO_2$ (2:1)), where the dispersed bright protrusions located on the Cu(111) lattice reflect Zn atomic positions in the same manner as for the UHV case (Fig. 1A).

When the temperature is raised to 373 K during NAP-STM imaging (Fig. 2A), the Cu terraces still appear atomically flat. At this level of magnification, the individual protrusions associated with the Zn atoms are not resolved. However, we can conclude that the introduction of the $CO_2/H_2$ mixture is not seen to induce any significant morphology

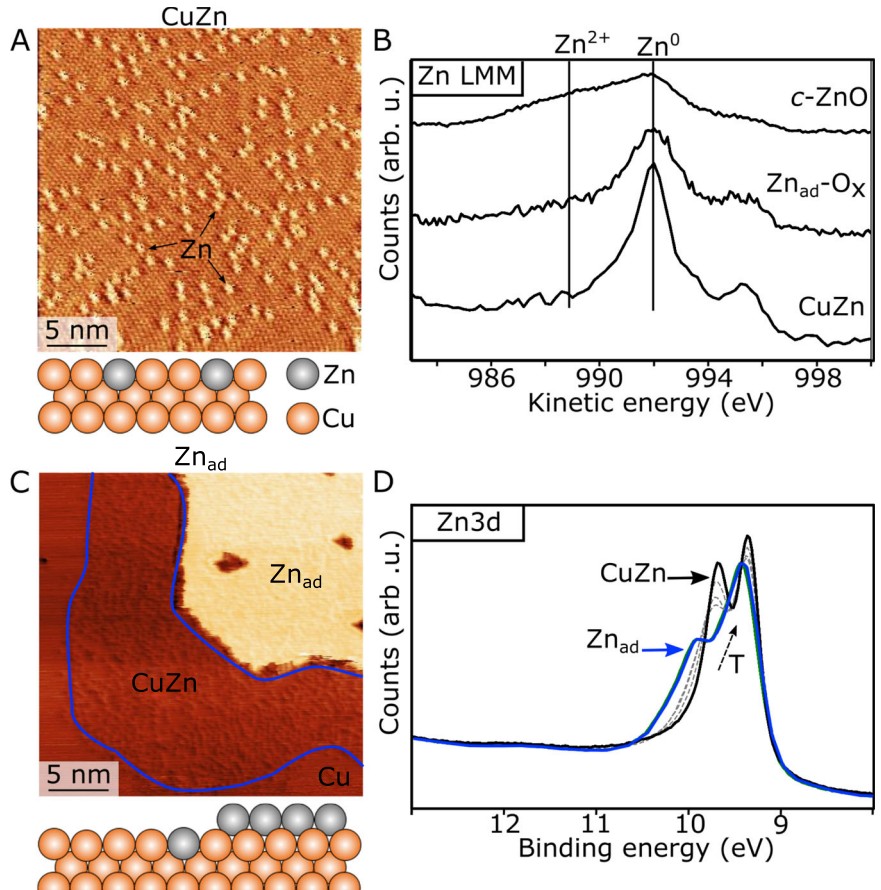

**Fig. 1 | The CuZn/Cu(111) alloy. A** Atom-resolved STM image of the CuZn surface alloy imaged in UHV. Zn atoms are substitutionally alloyed at Cu(111) sites ($Zn_{Cu}$) and appear as bright protrusions. The STM contrast of Zn species in the Cu(111) surface is reported in ref. 57. STM imaging conditions: $U_t = 66$ mV, $I_t = 0.31$ nA. **B** Zn LMM Auger spectra (hv = 1100 eV) recorded for the pristine CuZn surface and following oxidation in UHV at room temperature (monolayer $Zn_{ad}$-$O_x$ phase) and 480 K (monolayer, crystalline ZnO (c-ZnO)), respectively. **C** STM image of a

monolayer Zn island ($Zn_{ad}$) nucleated on Cu(111) following room-temperature deposition. The preferential exchange of Zn atoms from Zn island edge leads to CuZn alloy formation zone (blue outline) around the perimeter of the island. $U_t = 41$ mV, $I_t = 0.76$ nA. **D** Zn3d valence band (VB) photoemission spectra (hv = 85 eV) recorded in a temperature series from 300-480 K (T arrow) following the conversion of $Zn_{ad}$ islands into the CuZn alloy. Ball models reflect the surface structure schematically (sideview) (Zn: gray, Cu: orange).

changes, indicative of Zn segregation or ZnO formation (see next section). The main effect of the elevated temperature is the activation of mobility of Cu[60], evidenced by slightly reshaped edges and the formation of a few isolated $Cu_{ad}$ islands on the (111) terrace. The $Cu_{ad}$ islands have an apparent height of $2.1 \pm 0.1$ Å consistent with a Cu(111) step (see also Supplementary Fig. 2). This situation is maintained at the subsequent temperature point of 423 K in $H_2/CO_2$ (Fig. 2B). Here an even more prominent reshaping of the Cu surface seen by the presence of several monoatomic Cu(111) step heights is observed (indicated by numbers). Formate (HCOO) species were previously seen to induce strongly increased Cu diffusivity[61–63], and since the formation of HCOO from $CO_2$ hydrogenation is reported to be activated on Cu(111) already at room temperature[64,65], it is a likely cause of the accelerated Cu surface reshaping. Moreover, pronounced surface restructuring is also observed via UHV-STM acquired after the $H_2/CO_2$ gas exposure (Supplementary Fig. 3B). The observed surface roughening is likely driven by surface mobility of Cu, and we speculate that either entropy effects or preferential stabilization of step edges by $CO_2$ or HCOO adsorption could provide a driving force. For comparison, we also investigated the CuZn/Cu(111) phase in pure $CO_2$ at similar partial pressures and temperatures, where we again did not observe Zn segregation and ZnO formation (see Supplementary Fig. 4). Correspondingly, the reshaping of the Cu step edges was much less pronounced, possibly explained by reduced Cu diffusion in the absence of HCOO in

pure $CO_2$. The conclusion is therefore that $CO_2$ hydrogenation conditions induce a significant surface mobility of the Cu, but once formed, the CuZn remains as an intermixed phase with Zn in a metallic state. The relatively high coordination of Zn atoms in the Cu(111) surface compared to that on stepped surfaces or a nanoparticle could in principle present a kinetic barrier for Zn abstraction and oxidation. We exclude this as a significant cause of the apparent stability in the $H_2/CO_2$ gas since step edges are generated due to the mobility of the Cu surface at the elevated temperature, indicating sufficient mobility. Moreover, since the CuZn is stable in $CO_2$ without $H_2$ gas, it also seems that fast reduction of surface O species by the $H_2$ is not a dominant effect.

The stability of the CuZn alloy in $H_2/CO_2$ is confirmed by analysis of the Auger LMM peak recorded on the same sample. This peak has previously been used to discriminate between Zn in the oxidized and metallic state, due to a shift in the kinetic energy in the spectrum for $Zn^{2+}$ relative to $Zn^0$ (metallic)[66,67]. Accordingly, in Fig. 2D we do not see the emergence of a new peak in the Zn LMM spectra or a change in the Zn peak intensity for a sample analyzed after the gas exposure in NAP-STM experiments. This observation is in full agreement with the stable metallic state of the CuZn in both $CO_2$ and $H_2/CO_2$. We note that the corresponding O1s (Supplementary Fig. 7A) revealed the presence of some surface O species after $CO_2$ and $H/CO_2$ exposure, but the main peak position (near 530.9 eV) suggests the presence of O adsorbates

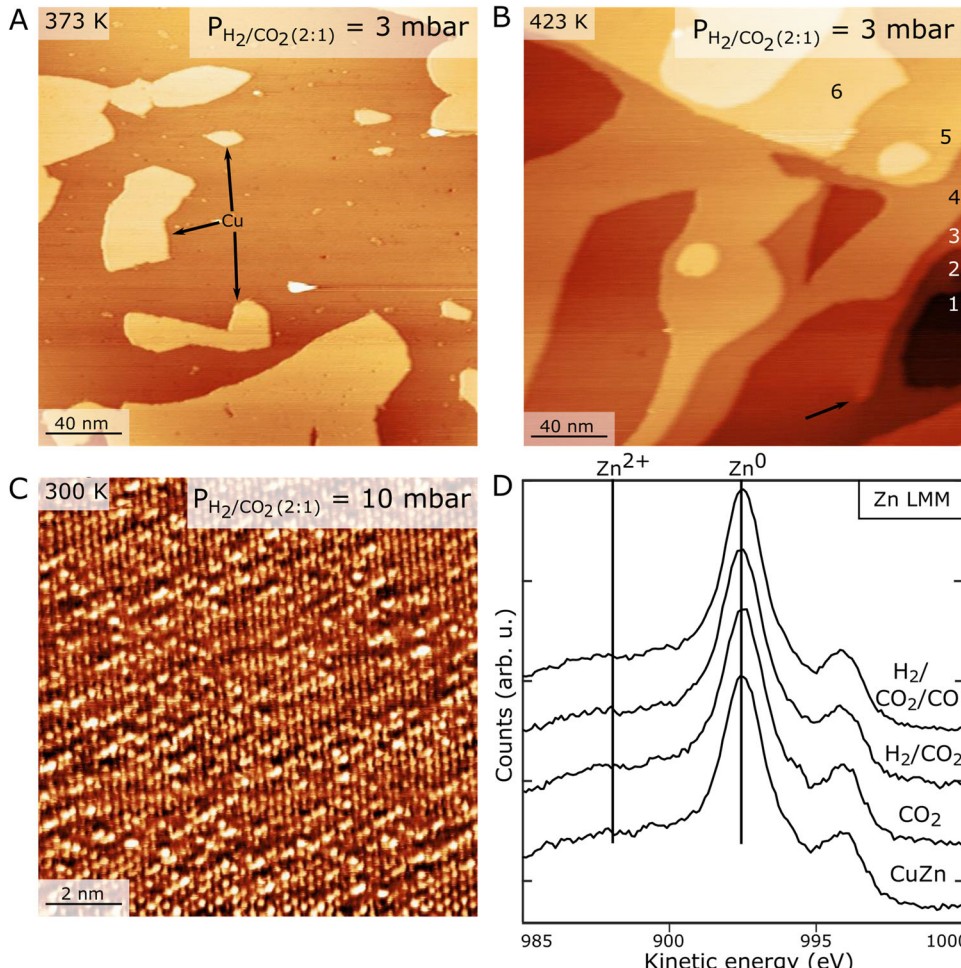

**Fig. 2 | Near-Ambient pressure STM images of CuZn/Cu(111) in $H_2/CO_2$ (2:1) gas.** **A** NAP-STM image in 3 mbar $H_2/CO_2$ gas recorded at elevated temperature of 373 K and (**B**) at 423 K. The numbering indicates the stepped structure of the CuZn/Cu(111) at the high-temperature conditions black arrow highlights a screw dislocation. STM imaging conditions: $U_t = 245$ mV, $I_t = 0.42$ nA and: $U_t = 312$ mV, $I_t = 0.31$ nA. **C** Atom-resolved NAP-STM image recorded at room temperature of the CuZn surface in 10 mbar (2:1) $H_2/CO_2$ gas. The bright protrusions reflect Zn atoms in the Cu(111) lattice. STM imaging conditions: $U_t = 14$ mV, $I_t = 2.17$ nA. **D** Zn LMM Auger spectra acquired in UHV before (CuZn) and after the 1.5 mbar gas exposure of $CO_2$ and 3 mbar gas exposures of $H_2/CO_2$ (2:1) along with $H_2/CO_2/CO$ (1:1:1). Spectroscopic signatures of oxidized ($Zn^{2+}$) and metallic ($Zn^0$) state of Zn are further indicated.

($O_{ad}$) on the Cu from residual gas absorption during spectrum recording rather than O bonded to Zn. This is in line with presence of metallic Zn species.

## Synthesis Gas Conditions ($CO_2/CO/H_2$)

The CuZn/Cu(111) surface morphology evolves in a remarkably different manner when it is exposed to a syngas mixture containing CO as an additional component, implying destabilization of the CuZn alloy. The NAP-STM experiments are again performed starting with a pristine CuZn/Cu(111) alloy with Zn atoms in the surface layer (Fig. 1A). This sample is then exposed to a pre-mixed syngas composition (1:1:1) $H_2/CO_2/CO$ (3 mbar) and then imaged in-situ at selected temperatures (Fig. 3A, B). At 300 K, no apparent changes in the surface structure of the CuZn/Cu(111) system were observed compared to the as-prepared sample. The surface morphology, however, changes distinctly when the sample temperature is elevated to 388 K in the syngas mixture (Fig. 3A). Here we now observe the emergence of a new phase (blue outline), which strongly resembles the monolayer $Zn_{ad}$ islands with the surrounding CuZn alloy phase, as illustrated in Fig. 1C. The images remarkably imply that a significant part of the Zn, initially present as alloyed Zn atom species in the surface layer, has segregated onto the surface in a

process driven by CO. The Zn island coverage (~0.08 ML in Fig. 3A) was always low, and never higher, than the initially deposited amount of Zn (~0.15 ML). This is consistent with a process where Zn segregates out and nucleates and grows into the observed islands. We note that the Zn assembles into rather large islands which affects the precision in the estimate of the average Zn coverage from available NAP-STM images recorded at these conditions, explaining why not all Zn contained in the initial alloy is accounted for.

When the imaging temperature is increased further to 423 K (Fig. 3B), we now also see the onset of roughening of the Cu(111) terraces as in $CO_2$ hydrogenation conditions (Fig. 2C), but as shown in the inserts, the monolayer islands can still be identified on the terraces of the stepped Cu surface. Segregation of Zn is also observed in NAP-STM studies of the CuZn/Cu(111) when the $CO_2$ component is removed from the reaction mixture corresponding to a premixed $H_2/CO$ (2:1) gas. The NAP-STM image in Supplementary Fig. 5 shows the morphology of the CuZn/Cu(111) in $H_2/CO$ at 386 K, where we again observe the formation of islands. It is clearly the addition of CO that leads to destabilization of the CuZn alloy, thus placing Zn atoms directly on the Cu(111) surface. This is underlined by the NAP-STM experiment in pure CO (Supplementary Fig. 6) showing that Zn segregation and agglomeration take place when the temperature is raised above room temperature.

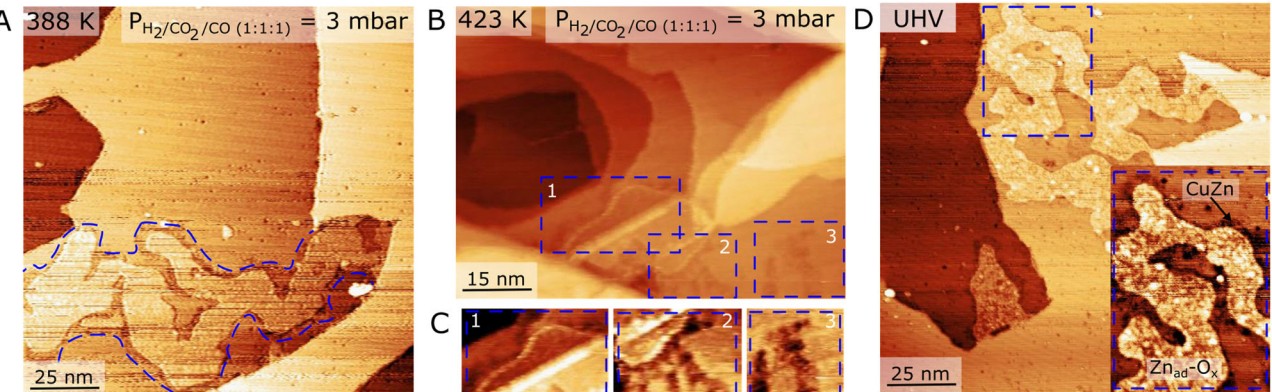

**Fig. 3 | Near-Ambient pressure STM images of CuZn/Cu(111) exposed to a H₂/ CO₂/CO (1:1:1) gas mixture. A** NAP-STM image was acquired after ca. 120 min. at 388 K in 3 mbar H₂/CO₂/CO gas. STM imaging conditions: $U_t = 153$ mV, $I_t = 1.12$ nA. The dashed blue line highlights the distinct Zn$_{ad}$/CuZn/Cu(111) interface region. **B** NAP-STM image acquired after ~160 min. in gas at 423 K in 3 mbar H₂/CO₂/CO gas. STM imaging conditions: $U_t = 346$ mV, $I_t = 0.38$ nA. **C** The images are high-contrast cut-outs from the STM image in (**B**) and highlight the presence of the Zn$_{ad}$ phase near the Cu step edges. **D** UHV-STM images acquired after the H₂/CO₂/CO gas exposure showing Zn$_{ad}$ structures surrounded by the darker-appearing CuZn interface region. The blue square marks the position of the inset. STM imaging conditions: $U_t = 3104$ mV, $I_t = 0.68$ nA.

Importantly, the post-analysis of the sample from the Auger LMM peak in Fig. 2C still reflects Zn in the Zn⁰ state, and thus not a Zn²⁺ oxide (see also Supplementary Fig. 7C for the corresponding LMM peaks recorded after pure CO and CO/H₂). The corresponding O1s and C1s XPS data recorded for the sample after NAP-STM in Supplementary Fig. 7A and 7B show no sign of C adsorbates, whereas the O1s indicates some level of O adsorption. The O1s peak is rather broad, and its direct assignment is influenced by O$_{ad}$ species on the Cu from background gas adsorption during the recording of the spectra. Nevertheless, the main peak is clearly shifted from the expected 530.3 eV for bulk ZnO on Cu (see also Fig. 4B,E). We note that the Zn3d peak, which is also sensitive to the formation of Zn$_{ad}$ (as shown in Fig. 1D using synchrotron XPS), was not resolved in high enough detail on the lab-source XPS connected to the NAP-STM.

The STM image contrast associated with the monolayer Zn$_{ad}$ phase formed from Zn segregation is seen to vary with the STM tip state both in NAP-STM images (Fig. 3A) and subsequent UHV imaging conditions (Fig. 3C). The monolayer Zn$_{ad}$ phase in NAP-STM appears significantly lower in height (1.3 Å) and less atomically smooth than the Zn$_{ad}$ islands in Fig. 1C, with a height modulation of ±0.3 Å (see Supplementary Fig. 2 and Supplementary Table 1). The apparent height in STM depends on the local density of states, which implies that the Zn$_{ad}$ phase may have an electronic character that is modified from as-deposited Zn, most likely due to O-related adsorbates. In STM images recorded after the gas atmosphere was removed, the morphology and height of the Zn$_{ad}$ islands is the same, but contrast variations stand out clearer (Fig. 3C). Here, the interior part of the island shows no distinct atomic periodicities, but its appearance is consistent with a Zn$_{ad}$ phase exposing Zn (bright) and O adsorbate covered parts (dark). The qualitative similarity and correspondence in height indicate that some O adsorbates are also present on the Zn$_{ad}$ islands during NAP-STM imaging (Fig. 3A).

A series of consecutive NAP-STM images recorded at 388 K after the image in Fig. 3A show that the size and morphology of the formed islands is maintained on a timescale of minutes, indicating that the CO-induced abstraction of Zn has reached a semi-steady state (Supplementary Fig. 8). The horizontal stripes arising from the STM scanning, particularly visible on the Zn island in the lower part of Fig. 3A, and slight modifications to the island perimeter in the image series (Supplementary Fig. 8), show that mass transport of Zn is taking place. Importantly, the interface between the Zn$_{ad}$ island perimeter and Cu(111) terraces in Fig. 3A-B is bordered by a distinct characteristic dark (~0.6 Å deep) and 10–50 Å wide region that directly follows the contour of the island (blue dashed outline). The same type of interface spontaneously forms when metallic Zn$_{ad}$ islands on Cu(111) are created directly by Zn deposition (see Fig. 1C and Supplementary Fig. 1). In previous STM work[57] this was shown to reflect a region with a high concentration of alloyed Zn atoms in CuZn. The region is created by intermixing of Cu and Zn atoms at the edge of a Zn$_{ad}$ island and the underlying Cu(111) surface, creating the sharp boundary between Cu(111), a CuZn zone and the Zn$_{ad}$ island (see Fig. 1C).

The NAP-STM image in Fig. 3C recorded after removal of the gas atmosphere and sample cooling shows that the CuZn interface is also created at the border of the Zn$_{ad}$ islands. Our interpretation of this is a dynamic situation present in NAP-STM syngas conditions where two processes are active and affecting the Zn. The first is the CO-induced segregation of Zn from the CuZn surface which supplies Zn to the Zn$_{ad}$ islands. Once the Zn$_{ad}$ phase is formed, a second process is established where Zn is supplied back to the Cu surface in the vicinity of the island edges due to favorable alloying. CO-induced surface segregation is a well-known phenomenon, evidenced on several binary alloys[68–70] including many Cu-based systems[71–73]. Greeley et al. also found in theory modeling that CO adsorbates lead to stabilization of Zn atoms in the surface of Cu(111)[74], but the direct extraction of Zn was not modeled. For CuZn/Cu(111) we therefore speculate that the destabilization of the CuZn surface alloy is driven by CO coordinating to Zn in the CuZn-surface, for example forming mobile Zn-carbonyl species. The CO-induced Zn segregation interestingly thus reduces the overall number of accessible Zn sites in the CuZn surface. The effect evidently contrasts with the behavior in CO₂ hydrogenation conditions in Fig. 2A-B, where the CuZn alloy surface was found to be stable.

## Zn segregation and oxidation

To investigate the mechanism of the dealloying process of CuZn we have conducted additional STM experiments for the CuZn alloy during gas exposure and high-resolution (synchrotron) XPS under controlled UHV conditions. Direct exposure of 10⁻⁷ mbar CO gas to pristine CuZn alloy in Fig. 1A does not induce Zn segregation. The STM images following CO gas exposure from 300 K to 550 K still reflect a stable alloy phase (Supplementary Fig. 9), whereas NAP-STM in pure CO at a similar temperature clearly leads to Zn segregation (Supplementary Fig. 6). This points to a mechanism of Zn segregation that depends on the CO partial pressure. Instead, we find that introduction of very mildly oxidizing conditions to the CuZn system at room temperature leads to Zn segregation and mild oxidation into

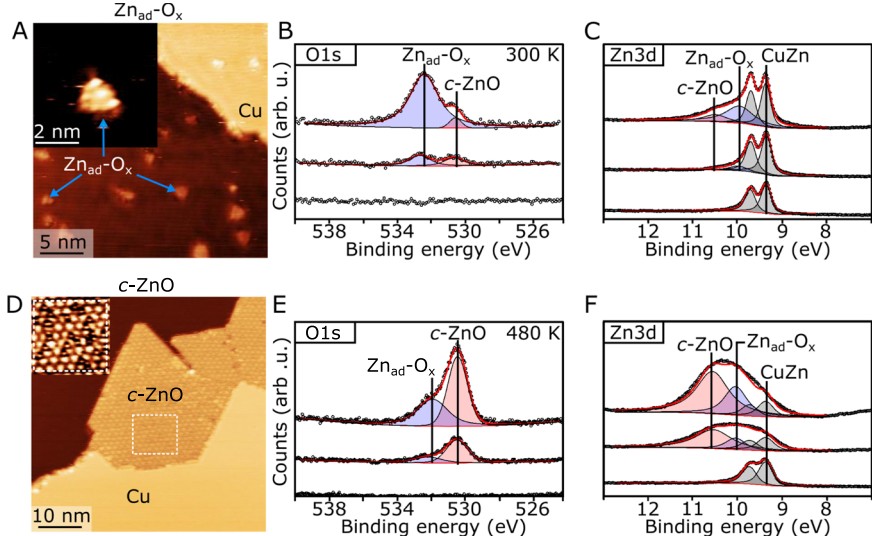

**Fig. 4 | Oxygen-containing phases formed from the CuZn/Cu(111) alloy. A** UHV-STM image of the $Zn_{ad}$-$O_x$ clusters formed by oxidation at room temperature (see also Fig. 3). STM imaging conditions $U_t$ = 443 mV, $I_t$ = 0.43 nA. **B, C** *O1s* XPS spectra (hv = 610 eV) and *Zn3d* valence band spectra (hv = 85 eV) showing the evolution of the peak structure for three steps during formation of $Zn_{ad}$-$O_x$ (0, 220 and 660 L $O_2$). **D** UHV-STM image of the monolayer crystalline c-ZnO phase formed by

oxidation of CuZn at 480 K. STM scanning conditions $U_t$ = 978 mV, $I_t$ = 0.22 nA. The insert (3.5 nm × 3.5 nm) shows a high-resolution STM image of the moiré structure for c-ZnO/Cu(111). **E, F** O1s XPS spectra (hv = 610 eV) and *Zn3d* valence band spectra (hv = 85 eV) for c-ZnO (0, 195 and 300 L $O_2$). Peak color shading: $Zn_{ad}$-$O_x$ (blue), c-ZnO (red) and CuZn (gray).

structures, which fits with the spectroscopic and structural signature of the $Zn_{ad}$ phase and nature of O species in this phase observed in-situ by NAP-STM.

Figure 4 A-B first illustrates an STM image and the corresponding *O1s* XPS spectrum for a CuZn/Cu(111) exposed to a small dose of pure $O_2$ at a pressure of $10^{-7}$ mbar at room temperature. Already at a $O_2$ dose of a 20 L at this temperature (1 L = $10^{-6}$ mbar × 1 s), the surface morphology has changed into a structure consisting of few nanometer wide Zn oxide clusters located on the Cu terraces, which denote as a $Zn_{ad}$-$O_x$ phase. The apparent height was again observed to be influenced by the STM tip state (Supplementary Fig. 10). In the most common imaging mode in Fig. 4A the height was determined to be 1.1 ± 0.2 Å relative to the Cu (see also line scans in Supplementary Fig. 2). This apparent height is consistent with the height of the more extended $Zn_{ad}$ islands observed in NAP-STM (Fig. 3A, C).

In the corresponding *O1s* XPS spectrum recorded for subsequent $O_2$ doses in Fig. 4B, a principal component around 532.2 eV (blue peak) is seen to form as $O_2$ is dosed. Moreover, a change in the chemical state of Zn is also evident from the *Zn3d* valence band signature (Fig. 4C) during $O_2$ exposure. The change is reflected by the emergence of a *Zn3d* peak shoulder (blue peak) on the high binding energy side as a function of $O_2$ dosage, which emerges as the CuZn alloy undergoes partial oxidation. The corresponding $O_2$ exposure performed at an elevated sample temperature of 480 K leads to faster agglomeration of Zn, but with the formation of a structurally different oxidized Zn structure, consisting of large ZnO islands which preferentially grow at Cu(111) step edges (Fig. 4D). The ZnO appears as fully crystalline monolayer islands (termed c-ZnO) with a 1.2 ± 0.1 Å apparent height in the STM images. Its structure is composed of a clear hexagonal lattice with an interatomic distance of 3.2 Å, and with a superimposed ~12 Å moiré lattice originating from the lattice mismatch with Cu(111). This appearance is in agreement with previously reported crystalline monolayer ZnO grown onto the Cu(111) surfaces[41]. Importantly, both the *O1s* peak shape and *Zn3d* peak structures are directly sensitive to the type of Zn-oxide formed on the Cu(111). In Fig. 4B, E, the *O1s* peak structure is fitted with two components at a binding energy of 530.3 eV (red peak) and 532.2 eV (blue peak), respectively. We find a direct correspondence between the high

(blue)/low (red) energy O1s component and the mildly oxidized Zn ($Zn_{ad}$-$O_x$) (Fig. 4B) and the c-ZnO phases (Fig. 4D), respectively. Moreover, in Fig. 1A, the peak at lower kinetic energies in the corresponding Zn LMM peak associated with $Zn^{2+}$ formation[52,75] is only observed for c-ZnO phase, suggesting that $Zn_{ad}$-$O_x$ differs fundamentally from fully oxidized Zn in ZnO.

In the *Zn3d* valence band spectrum, only the c-ZnO phase (Fig. 4F) is associated with a substantial change in the *Zn3d* peak structure compared to the CuZn, reflected by the new principal component at 10.6 eV (red). The shift in the *Zn3d* spectrum for $Zn_{ad}$-$O_x$ relative to the alloy is much less pronounced and is mainly seen as a broadening towards higher binding energies, which could be accommodated by a fitted peak at 10.1 eV (blue), in addition to the doublet peak from the Zn species in the CuZn. In Fig. 1C, the *Zn3d* spectrum for the $Zn_{ad}$ islands presented a similar broadening due to Zn-Zn hybridization (and therefore *d*-band formation like in Fig. 1D). The *Zn3d* spectrum is therefore directly showing that a significant degree of Zn-Zn-hybridization appears to be maintained in the $Zn_{ad}$-$O_x$ phase. This implies that it is not ZnO but instead it reflects a monolayer $Zn_{ad}$ island with co-adsorbed O species. The broad *O1s* spectrum recorded after $H_2$/CO/$CO_2$ NAP-STM (Supplementary Fig. 8A) contains several O species which complicates the assignment, but we note that the broad peak structure is consistent with the presence of some oxidized Zn-$O_x$ at the high binding energy position.

The combination of NAP-STM and experiments conducted under controlled oxidation under vacuum conditions thus proposes a link between a reduced $Zn_{ad}$-$O_x$ phase on Cu(111) with the segregated Zn phase observed in syngas conditions in Fig. 3. While $CO_2$ is evidently not directly able to oxidize the Zn in the CuZn alloy in the NAP-STM experiment (Fig. 2), we expect that some oxidation of the Zn phase from reaction with $CO_2$ (or reaction intermediates) may still occur. Partial oxidation of the Zn may occur when reactive Zn species are segregated by CO and released onto the surface in the CO/$CO_2$/$H_2$ gas mixture, either by reacting with $CO_2$ or reaction intermediates such as HCOO[74,76]. The presence of partially oxidized Zn on Cu(111), which deviates in structure and composition from bulk ZnO, has indeed been modeled theoretically[28,77–79], but mostly in the form of clusters and not the more coherent islands observed here.

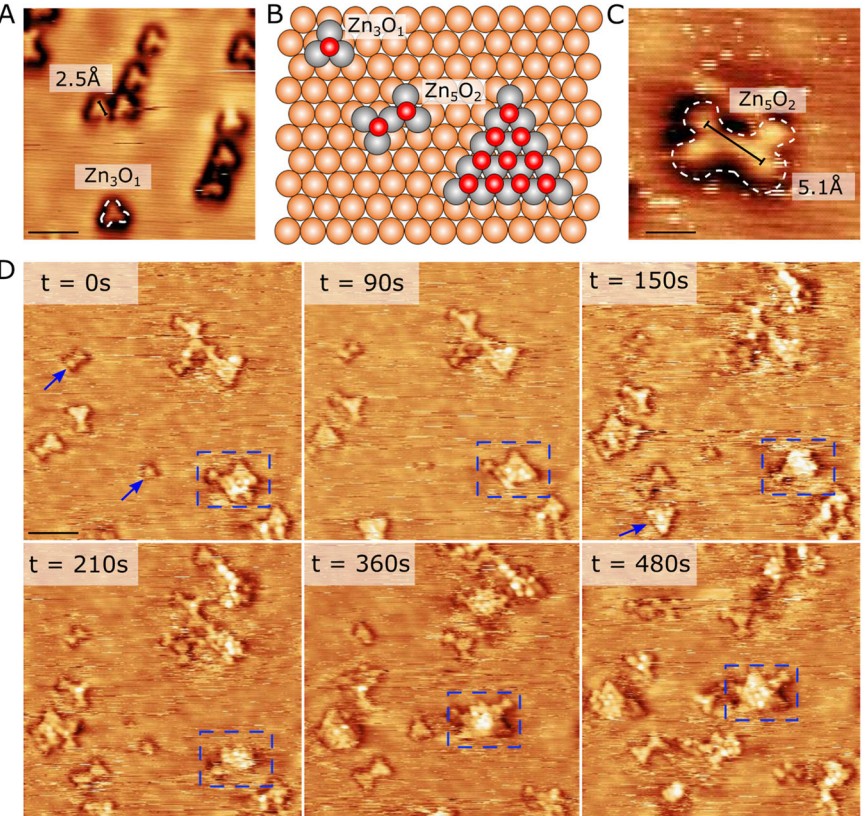

**Fig. 5 | Structure of the $Zn_{ad}$-$O_x$ clusters. A, C** UHV-STM images of $Zn_3O_1$ trimer clusters and a $Zn_5O_2$ cluster formed after 20 L $O_2$ dosing. STM scanning conditions $U_t = 12$ mV, $I_t = 0.37$ nA. and $U_t = 0.263$ mV, $I_t = 0.4$ nA. **B** Ball models (top view) illustrating the proposed structure of two primary $Zn_{ad}$-$O_x$ motif structures on Cu(111). Colors: Zn (gray), O (red), Cu (orange). **D** A sequence of six STM images of the same section recorded in-situ in $2 \times 10^{-9}$ mbar $O_2$ illustrating the gradual Zn segregation and agglomeration of $Zn_{ad}$-$O_x$ clusters. The timestamp (in seconds) refers to the first image when the STM resolution was stabilized which occurred some time after $O_2$ exposure commenced. STM scanning conditions $U_t = 1260$ mV, $I_t = 0.21$ nA. Blue arrows point to $Zn_3O_1$, $Zn_5O_2$, and a triangular cluster motif respectively, the structure of the larger clusters in the images can all be composed by merging the former two basic motifs.

## Dynamics of Zn oxidation

The dealloying of Zn from the CuZn layer was monitored in greater detail during $O_2$ dosing by time-resolved STM in Fig. 5. Upon the first introduction of 20 L of $O_2$ gas at $2 \times 10^{-9}$ mbar and room temperature (Fig. 5A), we see the emergence of distinct trimer-like cluster motifs that are clearly induced by the $O_2$ gas and formed on the Cu(111) surface. These trimers represent the smallest stable entities in the ensemble of features observed on the Cu surface. They are furthermore all aligned in the same high symmetry directions of the underlying Cu(111) hexagonal lattice and adopt the ~2.5 Å interatomic spacing of the Cu substrate. From their size and geometry and by comparison to the structures specifically modeled by recent density functional theory (DFT) work[78,80], a $Zn_3$-O trimer cluster composed of 3 Zn atoms stacked epitaxially on the Cu lattice and bound together with an O atoms appears as a candidate for the Zn trimers (Fig. 5B). Based on this primary motif, larger cluster types, such as the bowtie ($Zn_5O_2$) structure[78] also observed in the STM images (Fig. 5C) can be composed by addition of Zn and O.

The STM sequence in Fig. 5D depicts a selected area of the CuZn surface showing how larger $Zn_{ad}$-$O_x$ island structures form by continued $O_2$ dosing. The trimer and bowtie motifs can be recognized in the first frames of Fig. 5D (blue arrows), whereafter they in the subsequent frames grow by the addition of Zn or merge. The merging leads to larger clusters with a semi-ordered triangular shape (blue dashed square), that are still aligned with the underlying Cu lattice. Evidently, significant mobility of the Zn species in the alloy is activated already at room temperature upon exposure to $O_2$, leading to Zn segregation, diffusion, and agglomeration into large monolayer $Zn_{ad}$-$O_x$

islands (Fig. 5B). We also observe dark pits in the Cu surface both in these STM data and in the NAP-STM experiment (Supplementary Fig. 11 and Supplementary Fig. 12), which we associate with agglomerated Cu vacancies created during Zn abstraction. Importantly, the observation that the $Zn_{ad}$-$O_x$ clusters adopt a fixed homoepitaxial orientation on Cu(111) and the ~2.5 Å interatomic distance inherited from the substrate (or multiples hereof) (Fig. 5A and Fig. 5C), much shorter than any Zn-Zn distance in bulk ZnO (wurtzite), again demonstrates that the $Zn_{ad}$-$O_x$ phase (x < 1) is not related to a bulk form of ZnO. Instead, we conclude that $Zn_{ad}$-$O_x$ is composed of an epitaxial $Zn_{ad}$ layer with O adsorbates built from the units shown in Fig. 5B. This type of $Zn_{ad}$-$O_x$ phase agrees precisely with the partially maintained Zn-Zn hybridization concluded from the Zn3d valence band spectrum (Fig. 4C), and that Zn is not fully oxidized into $Zn^{2+}$ (Fig. 1B).

We rule out that residual $O_2$ impurities in either of the purified CO, $H_2$ or $CO_2$ gases would have influenced our NAP-STM data, since the Zn segregation in the NAP-STM experiment is activated only above room temperature. The CuZn phase never changed during exposure at mbar pressure of these gases at room temperature for the duration of the experiments (typically >200 min.). In contrast, the vacuum experiment in Fig. 5 clearly shows that room temperature exposure of only $10^{-7}$ mbar $O_2$, (corresponding to <100 ppb in 1 mbar), would have induced clear evidence of Zn segregation within minutes.

In the NAP-STM experiment, O adsorbates (e.g., O or OH) on the Zn may instead form due to oxygen-containing components present in methanol synthesis conditions, for example by reaction with $CO_2$, $H_2O$ (a product) or intermediates such as HCOO[34,81]. The CO and $H_2$ components in the gas imply overall strongly reducing conditions that

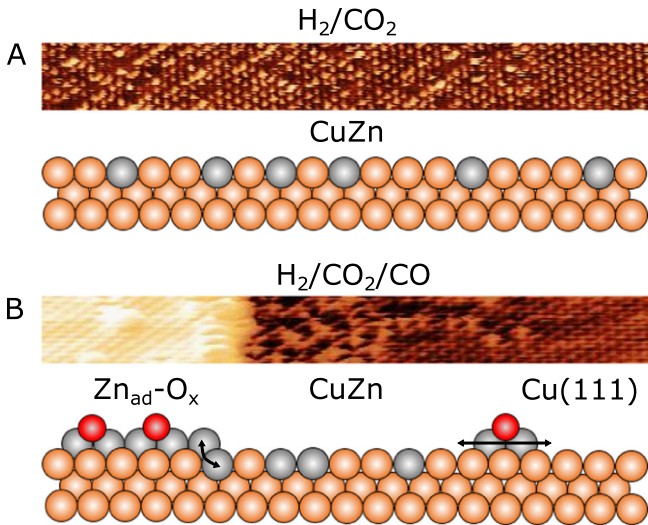

**Fig. 6 | Model of the CuZn/Cu(111) surface in H$_2$/CO$_2$ and in H$_2$/CO$_2$/CO.** (Zn: gray, Cu: orange, O: dark red) (**A**) atom-resolved NAP-STM image acquired of the CuZn/Cu(111) surface during the 10 mbar (2:1) H$_2$/CO$_2$ gas exposure. The structural state of the surface is represented in the ball model illustrating the stable CuZn alloy phase. (**B**) Representative atom-resolved UHV-STM image of the Zn$_{ad}$/Cu(111) interface region, showing the substitutionally alloyed Zn atoms near the edge of the Zn$_{ad}$ structure. The ball model illustrates the dynamic Zn$_{ad}$-O$_x$/CuZn/Cu(111) interface region emerging in H$_2$/CO$_2$/CO, arrows indicate the Zn alloying occurring in the Zn$_{ad}$-O$_x$/CuZn/Cu(111) interface region and the ongoing supply of the Zn$_{ad}$-O$_x$ clusters.

have a reducing effect on ZnO, explaining why the metallic-like Zn$_{ad}$-O$_x$ phase is formed rather than crystalline ZnO. The process observed in Fig. 5D may thus be expected to be accelerated in the NAP-STM experiment (Fig. 3) where the Zn$_{ad}$-O$_x$ has agglomerated into wide islands.

## Discussion

We used variable-temperature ambient pressure STM imaging of a well-defined CuZn/Cu(111) surface in CO, CO$_2$ and H$_2$ gas mixtures at mbar pressure to observe that its surface structure and chemical state is highly sensitive to the gas compositions. We show that the CuZn alloy surface is stable in mbar pressures of CO$_2$/H$_2$ (Fig. 6A), but when CO is added to the gas mixture, representative for methanol synthesis in syngas, a new surface state is formed. Zn is segregated onto the Cu surface and agglomerated in a Zn-O$_x$ type (x < 1) monolayer that co-exists with the CuZn alloy, which is then only present in a narrow brim zone around the Zn-O$_x$ phase. We interpret this as a dynamic process occurring at elevated temperature, where CO on the one hand induces fast Zn segregation and formation of diffusive Zn complexes which assemble into monolayer islands covered with O adsorbates (Zn$_{ad}$-O$_x$).

On the other hand, the Zn in the formed Zn$_{ad}$-O$_x$ phase is not bonded strongly as in ZnO and therefore sufficiently accessible, so that free Zn atoms can be exchanged back into the Cu at its perimeter, meaning that a new equilibrium is established where Zn-O$_x$ and CuZn phases co-exist the boundary (Fig. 6B). The exact distribution of these phases will be dependent on the CO pressure and the activation energies for the CO-induced Zn abstraction and insertion back into Cu. In the extreme case of no CO partial pressure, the CuZn surface is fully maintained in our experiment.

Previous TEM work shows that significant gas-induced morphology changes occur already at mbar pressure conditions[30,31], like those used here in NAP-STM, and that such changes could be correlated to reactivity changes. We, therefore, expect that the created interface type is likely to exist also in the active Cu/ZnO methanol catalyst in syngas, where the same type of dynamic interface between reduced Zn-O$_x$ and CuZn is established locally on the facets of Cu particles. Importantly, our spectroscopic measurements clearly show that Zn oxide can exist on the Cu(111) surface in both an oxidized crystalline state (Zn$^{2+}$O) and a reduced O-covered epitaxial Zn phase (Zn-O$_x$) with a close resemblance to a Zn film, whereof only the latter is present in the in-situ images. The reduced Zn-O$_x$ phase seems to be essential for atom exchange into a CuZn phase, whereas Zn in oxidized ZnO is expected to be more strongly bound[75,82].

The NAP-STM results presented here furthermore agree with recent NAP-XPS studies of a CuZn/Cu(211) surface at elevated pressure by Amann[8], concluding that the amount of CuZn correlates with a reduced form of on-surface Zn (Zn$^{\delta+}$) in CO/CO$_2$/H$_2$ gas similar to the the Zn$_{ad}$-O$_x$ phase. We note that the CO$_2$ partial pressure, which is 2 orders of magnitude higher than here and 5 orders of magnitude higher in the industrial catalysts, could drive the further oxidation of some of the Zn species into bulk ZnO, which is absent in our studies but observed in XPS in CO/CO$_2$/H$_2$. Therefore, a description of the active catalyst as a multiphasic system with ZnO, reduced Zn$_{ad}$O$_x$ (Zn$^{\delta+}$) and CuZn is in line with our observation, where the reduced and metallic Zn species are predominant under our conditions.

In the same NAP-XPS study, ZnO and Zn$^{\delta+}$ species were detected on the surface observed even in pure CO$_2$/H$_2$, which is in apparent disagreement with our observation of a stable CuZn surface in CO$_2$/H$_2$. However, this may be explained by the sequence in which CO was first dosed prior to the NAP-XPS observation[8] in pure CO$_2$/H$_2$, whereby segregation of Zn had already been activated, which could then be oxidized once on the surface.

Our observations may explain the apparent role of CO as both an inhibitor and promoter for methanol formation, and for transient phenomena that occur upon switching the CO content in the reaction atmosphere[16,31]. CO is needed in the activation of the pristine Cu/ZnO catalyst to reduce the ZnO in the first place and activate Zn migration, and in addition CO ensures the removal of far more severe inhibitors such as H$_2$O by the WGS reaction and adsorbed O species by CO oxidation. However, our study clearly shows that excess CO also drives a reduction of the exposed amount of CuZn. The destabilization of the alloy phase by the addition of CO to the gas-feed, may therefore explain why the activity of the CZA methanol catalyst decreases under low conversion conditions (-1 mol% of methanol in the effluent gas), where CO is less important for the removal of adsorbed H$_2$O[19].

Importantly, our results also suggest that the few-atom ZnO$_x$ clusters of the type imaged in Fig. 5A-C are present on the Cu surface in catalytic conditions when CO is used as part of the gas feed. Such clusters, whose concentration on the surface will be directly proportional to the amount of exposed CuZn, will be reaction-driven[83] and have previously been proposed in theory work to act as the active site for methanol formation[80,84]. The in-situ visualization of the CuZn system performed with STM thus generally agrees with a dynamic and complex multiphasic state of the operating Cu/ZnO catalyst, where the catalytic surface composition and surface Zn species dynamically adjusts to the gas environment and reaction conditions.

## Methods

### Near-Ambient pressure STM

The experiments were performed in an interconnected ultra-high vacuum (UHV) system consisting of separate chambers for sample preparation, analysis (laboratory-source XPS), and ambient pressure STM, respectively, all with a base pressure below 2×10$^{-10}$ mbar. The near-ambient pressure STM (SPECS Aarhus 150 NAP-STM) was housed on a separate side chamber separated by a gate valve and pumped by a turbopump and ion-getter pump, with facilities for gas admission to the STM cell. The system design allows us to transfer the sample between lab-source XPS and NAP-STM in UHV without exposing the sample to undefined gas conditions. We used etched W tips for STM

imaging. The bias voltage ($U_t$) refers to the voltage applied to the sample.

Ultrapure gases having a purity level of 5.5 N, 5 N, and 4.7 N for $CO_2$, $H_2$, and CO, respectively, were used. MicroTorr gas purifiers of the MC1 series were employed on all gas lines, facilitating an impurity level in the part per trillion range. Importantly, the gas purity of all gases was checked by performing extensive reference NAP-STM and post-XPS analysis experiments on clean Cu(111), Cu(110) and CuZn surfaces in mbar pressure of the pure gases, where oxidation due to residual $O_2$ was not detected from STM imaging and post XPS analysis within the time frame of these experiments (several hours). Gas mixing was done inside the NAP-STM cell by dosing the gases sequentially and always in the order of $CO_2$, CO and $H_2$. Mass spectrometry of the gas compositions was always employed during the experiment to ensure that the gases were properly mixed inside the reaction cell. The gas pressure was measured both at the inlet and outlet of the reaction cell by dual MKS Baratron gauges. The NAP-STM experiments were performed under static conditions with no exchange of the gases in the STM cell.

The heating of the sample inside the NAP-STM was carried out by radiative heating from a W filament located behind a sapphire window on the vacuum side of the NAP-STM cell. The maximum temperatures (423 K), the total pressure (3 mbar) and the applied $CO/CO_2$ and $H_2$ gas ratios used in this study were generally a compromise limited by the strong cooling effect of $H_2$ gas in the STM cell in relation to the maximum sample temperature. Variable temperature conditions were established while the STM tip was retracted by applying a heat ramp of ~1.5 K/min. requiring around 50 min to reach the desired set-point in the gas. The sample temperature was then stabilized, and the STM tip approached the surface for imaging.

### CuZn/Cu(111)

The synthesis of the CuZn planar model surface was carried in ultra-high vacuum (UHV). Cleaning of the Cu(111) single crystal substrate was performed by several cycles of argon sputtering (1 keV $Ar^+$-ions for 10 min) and annealing (810 K for 30 min) until the surface appeared clean in the STM images. For the Zn deposition step, a separate Zn evaporation cell (base pressure of $1\times10^{-9}$ mbar) attached to the main UHV system was employed to prevent Zn contamination in the main chambers upon deposition of Zn. Here, the evaporation was carried out by a home-built evaporator containing a Zn metal rod held in place by a W filament and a thermocouple connected to the lower end of the rod. The Zn coverage was adjusted by deposition time, using a pre-calibrated flux judged from Zn islands formed by Zn deposition at room temperature. The subsequent heating step was performed in the heating stage of the manipulator in the main UHV chamber.

### X-ray photoelectron spectroscopy

The UHV XPS/UPS experiments used for monitoring the Zn3d, O1s and some Auger LMM data (Figs. 1 and 4) were performed at the MATLINE beamline at the ASTRID2 synchrotron, Aarhus University. The beamline consists of a UHV chamber with a base pressure of $2\times10^{-10}$ mbar equipped with a Phoibos 150 electron energy analyzer, a SX700 monochromator, and an UHV-STM (Aarhus STM). Evaporation of Zn took place in the load-lock at MATLINE utilizing the same external Zn evaporation cell. The Zn coverage was estimated by calculating the relative area of metallic Zn films on the Cu(111) surface from STM images in combination with measuring the evolution of the Zn and Cu 3p levels by XPS. The oxidation of Zn took place by dosing oxygen gas through a leak-valve to a pressure of $2\times10^{-7}$ mbar, where the Zn/Cu(111) sample was kept either at 300 K or 480 K. The data acquisition was conducted at room temperature.

In addition, the analysis chamber connected to the NAP-STM included facilities for lab-source XPS (SPECS Phoibos 150 SCD) using an Al $K_\alpha$ X-ray source (SPECS XR 50). The XPS analysis was carried out on the as-prepared sample and after gas exposure by transferring the sample into and out of the ambient-pressure cell, all for samples kept at room temperature.

The peak fitting of the Zn3d was carried out based on the procedure presented in ref. 58. The three principal peaks attributed to CuZn (gray), $Zn_{ad}$-$O_x$ (blue) and c-ZnO (red) used for fitting the Zn3d spectra (Fig. 4) were fitted with a Doniach-Sunjic (DS) function capturing the Lorentzian (lifetime) and Gaussian (instrumental) broadening of the peak. The background was fitted by a Shirley function. The principal peaks in the O1s spectra were assigned to $Zn_{ad}$-$O_x$ (blue) and c-ZnO (red) based on the complementary information obtained via STM and further based on reference[85]. Peaks were again fitted by the DS function and the background was fitted by a linear function.

## Data availability
The data that support the findings of this study are available from the corresponding author upon request.

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

## Acknowledgements

Grants from the Independent Research Fund Denmark (Grant no. 9041-00070B) and Villumfonden (Grant no. 13264) are acknowledged (J.V.L.). Affiliations with the Center for Integrated Materials Research (iMAT) and the SMART Lighthouse at Aarhus University are acknowledged.

## Author contributions

Conceptualization of the project: J.V.L. NAP-STM: S.J., M.H. and L.L. performed the experiments and S.J. analyzed the STM data. XPS experiments: M.H.R.M., M.H. and S.J. performed the synchrotron measurements and M.H.R.M. analyzed the data. Equipment design: Z.L and L.L Supervision: J.V.L Writing – original draft: S.J and M.H.R.M. Writing – final draft, review, and editing: J.V.L with contributions from all co-authors.

## Competing interests

The authors declare no competing interests.
