## [Peer Review File · Nature Communications]

Visualizing the Gas-sensitive Structure of the CuZn Surface in Methanol Synthesis CatalysisREVIEWER COMMENTS

Reviewer #2 (Remarks to the Author):

Jensen et al. investigated the surface structure evolution of CuZn alloy species in CuZn/Cu (111) under different reaction atmosphere (CO₂/H₂, CO/CO₂/H₂) via variable-temperature/ pressure scanning tunneling microscopy. They found CO₂ and CO component displayed a distinct effect for CuZn structure. CO₂ with weak oxidation capacity have hardly obvious influence on CuZn phase. In comparison, CO can surprisingly destroy the CuZn stable structure and induce the separation of Zn specie to Cu (111) surface thus form partial-oxide Znad-Ox species. This research result is interesting and inspiring, which fills the blank of STM technology applied on active-site identification and dynamic evolution. However, I think several issues should be still considered and solved, specific as below.

1. Methanol synthesis is a structure-sensitive reaction for Cu catalysts. The manuscript just focuses on the evolution of structure of CuZn model catalyst under different reaction atmosphere, however, I am also interested in the corresponding catalytic results. How about the consistent of the reaction performance and surface chemistry study?
2. I strongly advise authors add data about catalytic performance, e.g., MS or IR connected with in-situ STM can be used to capture the vital active intermediates (HCOO*, CH₃O*...), which may be a good way to reflect the structure-activity relationship.
3. Typically, the feedings for methanol synthesis is CO/H₂=1:2 or CO₂/H₂=1:3. Why authors don't use the same condition to conduct the related experiment?
4. As known, the methanol synthesis from CO/CO₂ is usually at higher temperature and much higher pressure (200-250 oC, 2-5Mpa). Especially under real reaction pressure, the CuZn state is generally significant differences in comparison to its AP-STM/XPS model study (Nature Catalysis 2021, 4, 488–497). So, can the authors eliminate this impact or propose a convincing viewpoint that supports the relationship between CuZn catalysts structure and real methanol synthesis reaction.
5. In Figure 1C, I cannot find clear boundary between CuZn alloy and Cu substrate from contrast and color etc.. Even though I can imagine that CuZn alloy exist between Znad and Cu, whether there is strong evidence from other characterizations to confirm it.
6. Please note the detail in the text, according to the previous conclusion, the content of “Correspondingly, the reshaping of...due to HCOO is possible in pure CO₂” in line 164-165 should be revised to “due to HCOO is impossible in pure CO₂”, please check the full text for other- expression / writing errors, this may misunderstand the reader.
7. In Supplementary Figure 8, are there any evidences to support the fact of assigning 532.2 eV to Znad-Ox species? If so, please add the details.
8. Based on the author's understanding of CuZn catalysts and CO/CO₂ hydrogenation processes, can the authors provide more specific suggestions for the design of real catalysts for methanol synthesis? It will attract and guide more readers and researchers.

Reviewer #3 (Remarks to the Author):

The current paper by Jensen et al. attempts to experimentally describe the structural and chemical composition change of a model CuZn(111) alloy catalyst during methanol synthesis. The paper is nicely written, and all the data have obtained with carefully conducted experiments and with sound data interpretation. However, I cannot recommend publication in Nature Communication since the link to methanol synthesis conditions is not realistic. Real methanol synthesis takes place at pressures of around 50-100 bar and not at a few mbar in pressure. The current study is off by 4-5 orders of magnitude. In comparison with the Amann study where spectra were obtained at 2 orders of magnitude higher pressure and thereby claiming a disagreement becomes not relevant. For instance, could the flat (111) surface not to be significant reactive for low dosage of CO₂ at the current pressure and therefore the alloy surface becomes intact and could the low reduction potential of H₂ not be high enough to reduce the partial oxidized Zn Island in the CO₂/CO mixture. Approaching atmospheric condition, the results may become completely different making the study redundant as being related to methanol synthesis conditions.

The study is therefore more related to a traditional surface science investigation that would be better suited in a regular journal.

Reviewer #4 (Remarks to the Author):

Report on manuscript entitled "Visualizing the Gas-sensitive Structure of the CuZn Surface in Methanol Synthesis Catalysis" by Jensen et al.

The manuscript deals with the timely topic of the understanding of atomistic processes of methanol synthesis by CuZn based catalysts. The conclusions are drawn from in-situ near ambient pressure STM results and ex situ XPS, as well as UHV STM results. The experimental results are interesting, but there is some additional proof required that the conclusions drawn from the STM image contrast are correct or other possible scenarios need to be discussed. In addition, the XPS results were obtained under UHV conditions after the in-situ STM experiments, which may have altered the surface and are not directly comparable to the STM observations.

Further on, it is debatable that conclusions from an experiment in the mbar regime can be extrapolated to 200 bar pressure. In addition, industrial catalysts are composed of ZnO and Cu nanoparticles. The question arises to which extent studies of a planar model system are relevant for industrial catalysis. A rigorous comparison with published theory results is lacking, which could further substantiate the hypothesis proposed in this manuscript. One of my main concerns is that the introduced Znad-Ox phase lacks experimental evidence.

More detailed comments:

1. On page 4 the STM results after Zn deposition and annealing are described and bright features are

assigned to Zn atoms. How can the authors be sure that the model in Fig 6a is correct? Could this also be Zn adatoms or impurities from the deposition? What is the apparent height of these objects? The contrast appears to be much too high to be alloy related, compared to literature results. What are the vacuum conditions during Zn deposition and transfer? Zn is very reactive.

2. In a CO₂ / H₂ mixture (from page 5 on), the authors report the formation of Zn dimers, which find very hard to distinguish on the STM images from what the authors assign to single Zn atoms. Would such a phase separation be expected in this low concentration Zn regime from the bulk Cu-Zn phase diagram? How does the Zn 3d spectrum evolve (see Fig 1D), which is sensitive to the Zn distribution in the surface?

3. Under CO₂/H₂/CO mixtures (from page 8 on) the authors state that the images show that a significant amount of Zn has segregated to the surface (line 206). It became not clear so far, that the authors assume that Zn is also subsurface. Can they give some estimate (eg based on the amount deposited and seen by STM), how much Zn is subsurface after the sample preparation? The authors assume that this behavior is related to CO. Here a comparison with literature CO adsorption energies of Cu and Zn might strengthen the argument.

4. It is not clear, why the authors assign the new structures observed under CO₂/H₂/CO to a Znad-Ox phase. It is rather confusing that this terminology is introduced already from line 230 on, without giving any rational argument. Where is the oxygen coming from? The low pressure CO control experiment does not show the same behavior (page 10 on) put instead at CO pressures in the mbar regime Zn segregation is observed (Fig. S7). In line 274 the authors state that "While CO₂ is not directly able to induce Zn segregation and oxidize the CuZn alloy..." This is contradicting literature results, in which the oxidation of Cu nanoparticles by CO₂ was reported. The only evidence for oxygen comes from the O1 s results in Figure S8 A, which shows a higher signal after CO exposure as compared to after H₂/CO₂/CO exposure. To my opinion, it is therefore very speculative to assign the features observed in STM in CO₂/H₂/CO mixture with a Znad-Ox phase. The XPS spectra after low pressure oxidation shown in Fig 4B and C differ significantly from the XPS results shown in Fig. S8 obtained after H₂/CO₂/CO exposure, and there is no direct comparison with the Zn 3d levels available.

Reviewer #5 (Remarks to the Author):

In their ms "Visualizing the gas-sensitive structure ..." Jensen et al. present an STM investigation of a Cu/Zn surface alloy under synthesis gas mixtures. They show that, when the gas mixture contains CO in addition to H₂ and CO₂, the surface alloy partially de-mixes. Zn islands covered by adsorbed O atoms are formed that are surrounded by a broad rim, in which a fraction of the Zn atoms remain alloyed in the Cu surface. Control experiments with O₂ support the interpretation of the data. A claim is made that this configuration of ZnOx islands surrounded by a Zn/Cu alloy represents the active state of the Cu/ZnO/Al₂O₃ methanol catalyst.

As far as I can judge, the STM and the supporting XPS experiments have been carefully performed, and

the interpretation of the data is altogether conclusive. Methanol synthesis from synthesis gas is an important industrial process, and the active state of the unusual catalyst that combines a metal and a reducible oxide has remained an open question. There have been previous investigations that point into similar directions as the present study but the atomic resolution of the formation of the active state under synthesis gas and the analysis of the detailed morphology are new. The results are certainly interesting for the catalysis community. However, I have several questions on which the authors should comment:

Fig. 1D, black spectrum: The intensity ratio of the Zn 3d(5/2) : 3d(3/2) peaks seem to be lower than the expected 3:2. Please explain. (The tunneling voltage and current data are missing in Fig. 1C.)

Fig. 2A and B: What is the driving force of the morphology change? The presence of the H₂/CO₂ gas phase may explain an enhanced mobility, but it cannot constitute the driving force, a thermodynamic property, in contrast to what the authors suggest. The same holds for the discussion on page 10, top paragraph.

Fig. 2C: It appears that most of the Zn dimers point into the same direction, which may indicate a double tip, a frequent artifact in STM data. Whether such an artifact is present could easily be decided by determining the distribution of the orientational angles of the dimers. True dimers should display random orientations.

Page 6 and Fig. S4. A statement is made that CO₂ alone does not lead to dealloying. What does H₂ alone do?

Fig. 3A. The coverage of Zn islands was 0.08 +/- 0.03 ML. It is not a trivial matter to obtain good statistics of the coverages of structures that are similarly large as the diameters of the STM images. Please explain the method. How many images from different areas have been analyzed? Have automatic tools been used to determine the areas? From the area covered by Zn islands and the coverage of O atoms obtained from the O 1s peak in XPS one can evaluate the local coverage of O atoms on the islands. This might be an important parameter that discriminates the ZnO_x islands from stoichiometric ZnO. (The caption of Fig. 3D is missing; 3E is not marked.)

Have any experiments been performed to analyze whether the dealloying was reversible, i.e., did the islands disappear when the gas was changed from H₂/CO₂/CO to H₂/CO₂?

On page 10, top paragraph, it is argued that the Zn islands on the alloy rim around the islands are active. It is not clear to me on which observation this is based. Couldn't the islands themselves be active?

Altogether, I find the language and terminology somewhat inexact. On page 10 a statement is made that "... an equilibrium is established [...] in the vicinity of the island edges", and a similar statement is made in the abstract. The term equilibrium suggests a permanent exchange of Zn atoms between the edges and the islands, but this has not been observed. The term "AP-STM", used throughout the ms, is not correct. AP (ambient pressure) means 1000 mbar, not the more than two orders of magnitude lower pressures applied in the experiments. That the term has been used in previous work at similarly low pressures is not a sufficient argument. The claim made in the abstract that the work "... establishes a complex role of CO as an activator of catalytic activity for methanol formation and as an inhibitor ..." is not really based on the data because no activity has been measured in this work that could justify such a conclusion.

Reviewer #2:

1. Methanol synthesis is a structure-sensitive reaction for Cu catalysts. The manuscript just focuses on the evolution of structure of CuZn model catalyst under different reaction atmosphere, however, I am also interested in the corresponding catalytic results. How about the consistent of the reaction performance and surface chemistry study?

Answer: We thank the reviewer for the suggestion. In the paper we have focused on the structural and chemical properties of well-defined surfaces that reflect a CuZn alloy surface and ZnO_x/Cu. This catalyst system is the most used industrial catalyst for methanol synthesis, with a very well demonstrated catalytic activity for methanol synthesis. The main scientific questions addressed here are therefore concerned with an accurate description of the active surface, and especially the interconversion between ZnO_x and Zn species in the Cu(111) surface. The referee is correct, that it would be desirable with additional insight into the specific reactivity performance here, but experiments using scanning tunneling microscopy is often a compromise between structural insight and measuring actual catalytic performance. We refer to the paper by Amann et al. (Ref 8), who used ambient-pressure XPS on similarly prepared samples to detect product species, confirming that some level of methanol formation reactivity is obtained even for model studies, which is in line with other low-pressure surface science studies.

Action: On page 2, we now write “Moreover, a surface sensitive in-situ method based on ambient-pressure X-ray photoelectron spectroscopy (AP-XPS) was recently applied by Amann et al. to demonstrate that both oxide and alloy phases seem to co-exist on a Zn/ZnO/Cu(211) model catalyst, and that formation of relevant methoxy and formate reaction intermediates could be observed on this type of model system, in agreement with IR spectroscopy results (31,32)”.

Here we refer to two IR papers that established the validity of these kinds of model systems studies Surface Science 402-404, 92-95 (1998) and J. Chem. Phys. 152, 044703 (2020) .

2. I strongly advise authors add data about catalytic performance, e.g., MS or IR connected with in-situ STM can be used to capture the vital active intermediates (HCOO*, CH3O*...), which may be a good way to reflect the structure-activity relationship.

Answer: We agree with the referee that extensive information on the evolution of surface species would be desirable, but as mentioned in the reply to comment 1, this has been extensively investigated in several papers for comparable model systems, and it would also be a very significant additional series of experiments that need to be performed. What has been missing in the literature is an atomic and nanoscale characterization of the surface structure while it is exposed to reaction conditions, which is the main scope of our paper.

3. Typically, the feedings for methanol synthesis is CO/H₂=1:2 or CO₂/H₂=1:3. Why authors don't use the same condition to conduct the related experiment?

Answer: The reviewer is correct that industrial methanol synthesis applies these gas ratios. The experimental conditions applied here were defined to be able to compare in the best possible way with literature performed with varying gas compositions under model conditions (mbar) pressure such as those in Journal of Catalysis **262**, 65-72 (2009) and Science **376**, 603-608 (2022). In addition, we are limited by the maximum pressure, in particular of H₂, due to the strong cooling effect of H₂ which would limit the accessible temperatures. We do not expect that this difference in gas ratio will significantly change the main conclusions of the paper.

Action: On page 17-18 (methods) we have written: The maximum temperatures (423 K), the total pressure (3 mbar) and the applied CO/CO₂ and H₂ gas ratios used in this study were generally a compromise, limited by the strong cooling effect of H₂ gas in the STM cell in relation to the maximum sample temperature.

4. As known, the methanol synthesis from CO/CO₂ is usually at higher temperature and much higher pressure (200-250 °C, 2-5Mpa). Especially under real reaction pressure, the CuZn state is generally significant differences in comparison to its AP-STM/XPS model study (Nature Catalysis 2021, 4, 488–497). So, can the authors eliminate this impact or propose a convincing viewpoint that supports the relationship between CuZn catalysts structure and real methanol synthesis reaction.

Answer: This is an important point also noted by reviewer #3.

It is well accepted that detailed characterization of catalysts will be a compromise between using sophisticated methods which can be applied under simplified conditions and measuring at industrial conditions which strongly limit the accessible methods. The STM method is normally a vacuum technique with most publications of model catalysts performed under such conditions. STM is in principle not limited by the pressure, but elevated pressure studies with STM are in practice always done at reduced pressure (mbar range typically, with a few examples to 1 bar range) compared with real conditions. There are so far no methods available that have studied the surface of a catalyst operating at 20-50 bar. The paper referenced by the reviewer is a good example, since the authors use XAS which is distinctly non-sensitive to the topmost surface. That paper indeed shows that Zn changes chemical state as a function of pressure and gas composition, but it is not possible to directly conclude if the change of Zn state happens on the surface of Cu or elsewhere. This is where our results fill an important gap. The main finding is that the surface morphology responds very strongly to changes in the gas composition, in particular from the CO component, and we strongly believe that this observation will be relevant for the interpretation of catalyst changes under real conditions. To make it clear that we study the sample under lower nominal pressures, we have now emphasized this even more clearly in both introduction and conclusion sections.

Action: On page 3 we write: The STM technique is not fundamentally limited by the pressure, but for the practical application the accessible pressure range in NAP-STM has so far typically been from a few mbar to bar pressure, which is lower than the 50-100 bar applied in industrial methanol catalysis.

On page 16-17 (see also reply to reviewer #3) we specifically address the role of the lower pressure by writing: “We note that the CO₂ partial pressure, which is 2 orders of magnitude higher than here and 5 orders of magnitude higher in the industrial catalysts, could drive the further oxidation of some of the Zn species into bulk ZnO, which is absent in our studies but observed in XPS in CO/CO₂/H₂. Therefore, a description of the active interface as a multiphase interface between ZnO, reduced Zn_{ad}O_x and CuZn system is in line with our observation, where the most reduced species are predominant under our conditions.”

5. In Figure 1C, I cannot find clear boundary between CuZn alloy and Cu substrate from contrast and color etc. Even though I can imagine that CuZn alloy exist between Zn_{ad} and Cu, whether there is strong evidence from other characterizations to confirm it.

Answer: The boundary between Zn island, Cu(111) and the alloy phase results from dissolution of Zn into the topmost layer of Cu in the vicinity of the Zn island. The process is active at the Zn perimeter. As preparation for this study, and due to the significance of this process, we have carried out a full study reported in refence 55 (Mammen et al., “Atomic-Scale Site Characterization of Cu–Zn Exchange on Cu(111)”. The Journal of Physical Chemistry C 127, 3268-3275 (2023)), reporting on the dissolution

of Zn. The figure below shows the formation of the boundary, and the further analysis in that paper confirms the formation of the CuZn alloy and how such Zn atoms species are imaged in STM. We recognize now that this may not have been clearly explained and therefore, we have revised and added more information and a new figure S1 on this point.

Figure R1: Figure 1 from *The Journal of Physical Chemistry C* 127, 3268-3275 (2023).

Action: On page 4 we have added the following: “A previous study in Ref⁵⁵, has shown that slow intermixing of Zn into the topmost layer of the Cu(111) surface occurs near the Zn_{ad} island perimeter, leading to a sharp boundary between alloy and Cu, seen from the distinct darker STM contrast of the mixed CuZn zone highlighted in Fig. 1c compared with the bare Cu(111).”

6. Please note the detail in the text, according to the previous conclusion, the content of “Correspondingly, the reshaping of...due to HCOO is possible in pure CO₂” in line 164-165 should be revised to “due to HCOO is impossible in pure CO₂”, please check the full text for other- expression / writing errors, this may misunderstand the reader.

Answer and Action: The original sentence was possibly ok, with the wording ‘...no enhanced Cu diffusion due to HCOO is possible...’, but we agree that this can also be written in the way suggest by the reviewer, which we have now done.

7. In Supplementary Figure 8, are there any evidences to support the fact of assigning 532.2 eV to Zn_{ad}-O_x species? If so, please add the details.

Answer: The experiment reported in Figure 4 was done by oxidizing Zn at two different temperatures, and it indeed serves to show that Zn species on the surface can exist as an oxidized species with O1s component at 532.2 eV, which we link to the oxidized Zn clusters (interpreted as Zn_{ad}-O_x) in the STM figure generated a low temperature (Fig. 4A) and a significant different ZnO (like bulk) component at 530.3 eV generated by oxidizing a higher temperature. In both cases there is no oxidation of the Cu, so the significance is that oxidized Zn can exist without having crystallized as ZnO (bulk). Moreover, the bulk ZnO peak is associated with a shift in the Zn LMM spectrum, which is not seen when the O1s peak at 532.2 is present. The data reported in Supplementary figure 8 are recorded post-STM, i.e. after the transfer from the STM cell to the analysis chamber (through UHV). Here we have indicated the expected peak position the Zn_{ad}O_x component in the O1s spectrum, to make it visible that the broad O1s peak encompasses this component. We understand if this way of comparing the spectra was not evident from our first description, so we have revised the manuscript substantially to make this clear.

Action: On page 9 we have added: “Importantly, the post-analysis of the sample from the Auger LMM peak in Fig. 2C still reflects Zn in the Zn⁰ state, and thus not a Zn²⁺ oxide (see also

Supplementary Fig. 8C for the corresponding LMM peaks recorded after pure CO and CO/H₂). The corresponding O1s and C1s XPS data recorded for the sample after NAP-STM in Supplementary Fig. 8A and 8B show no sign of C adsorbates, whereas the O1s indicates some level of O adsorption. The O1s peak is rather broad, and its direct assignment is influenced by O_{ad} species on the Cu from background gas adsorption during the recording of the spectra. Nevertheless, the peak is clearly shifted from the expected 530.3 eV for bulk ZnO on Cu (see also Fig 4B,E), in line with the assignment of Zn_{ad} in a metallic state. We note that the Zn3d peak, which is also sensitive to the formation of Zn_{ad} (as shown in Fig. 1D using synchrotron XPS), was not resolved in high enough detail on the lab-source XPS connected to the NAP-STM.”

On page 11, we then revise the STM discussion in relation to this discussion: “The monolayer Zn_{ad} phase in NAP-STM appears significantly lower in height (1.3Å) and less atomically smooth than the Zn_{ad} islands in Fig. 1C, with a height modulation of ±0.3 Å (see Supplementary Fig. 2 and Supplementary Table 1). The apparent height in STM depends on the local density of states, which implies that the Zn_{ad} phase may have an electronic character that is modified from as-deposited Zn, most likely due to O-related adsorbates. In STM images recorded after the gas atmosphere was removed, the morphology and height of the Zn_{ad} islands height is the same, but contrast variations stand out clearer (Fig. 3C). Here, the interior part of the island shows no distinct atomic periodicities, but its appearance is consistent with a Zn_{ad} phase exposing Zn (bright) and O adsorbate covered parts (dark). The qualitative similarity and correspondence in height indicate that O adsorbates are also present on the Zn_{ad} islands during NAP-STM imaging (Fig. 3A).”

8. Based on the author's understanding of CuZn catalysts and CO/CO₂ hydrogenation processes, can the authors provide more specific suggestions for the design of real catalysts for methanol synthesis? It will attract and guide more readers and researchers.

Answer: The goal of these kinds of model studies is indeed in the end to help the formulation of new catalysts and/or improved operating conditions for methanol synthesis. Our research activity does not include the synthesis of catalysts, but we do think that the main findings of our paper will be well received, considering the many observation on how activation in different gases affects the activity in the short and long term. Ultimately, the insight here might help the development of methanol synthesis catalyst that can work with pure CO₂ and H₂, avoiding the CO component, since we demonstrate that the CuZn alloy is stable even in CO₂. At this point, it is however, too early to point to specific development strategies. We believe that the finding of CO-induced destabilization can eventually be used to understand how to activate the catalyst in the best possible way.

Reviewer #3:

The current paper by Jensen et al. attempts to experimentally describe the structural and chemical composition change of a model CuZn(111) alloy catalyst during methanol synthesis. The paper is nicely written, and all the data have obtained with carefully conducted experiments and with sound data interpretation. However, I cannot recommend publication in Nature Communication since the link to methanol synthesis conditions is not realistic. Real methanol synthesis takes place at pressures of around 50-100 bar and not at a few mbar in pressure. The current study is off by 4-5 orders of magnitude. In comparison with the Amann study where spectra were obtained at 2 orders of magnitude higher pressure and thereby claiming a disagreement becomes not relevant.

Answer: We thank the referee for the relevant comments. We reply to this in two parts as follows:

1. The referee is right that there is a 2 order gap between the experimental pressure used in our study and in the referenced paper by Amann. We would, however, like to point out that we are not claiming

disagreement with that paper, and we regret if our paper could be perceived in this manner. In contrast, our paper uses a comparable starting point (CuZn alloy) with the intention of providing new information on surface structure and distribution of Zn in the topmost layer of a CuZn alloy surface on Cu(111). We believe we have successfully achieved this. In both the Amann study and ours, a mobility of Zn from the alloy to the surface is observed, but our main new finding is that it is CO that accelerates this process. Therefore, once the system has seen CO, the Zn is moved onto the surface facilitating the formation of Zn-O species on the surface. The reviewer is correct that it is plausible that the higher pressure of CO₂ in that study, oxidizes some of the Zn_{ad} species further on to bulk-like ZnO, and we certainly do not want to claim a disagreement with this observation. An important point however, is that, in the Amann paper, the reported gas variation starts with CO/H₂ dosing with a subsequent exchange of CO to CO₂ in the subsequent spectra for the same sample. According to our work, the effect of CO-induced Zn segregation is thus already activated from the beginning and therefore a direct comparison with our observation of a stable CuZn surface in pure CO₂/H₂ experiment without any prior CO dosing is not possible. We have made this clear in our revised version.

Action: On page 16-17 we have explained our comparison with the NAP-XPS results more precisely: “The NAP-STM results presented here furthermore agree with recent NAP-XPS studies of a CuZn/Cu(211) surface at elevated pressure by Amann⁸, concluding that the amount of CuZn correlates with a reduced form of on-surface Zn (Zn^{δ+}) in CO/CO₂/H₂ gas, i.e., similar to the Zn_{ad}-O_x phase. We note that the CO₂ partial pressure, which is 2 orders of magnitude higher than here and 5 orders of magnitude higher in the industrial catalysts, could drive the further oxidation of some of the Zn species into bulk ZnO, which is absent in our studies but observed in XPS in CO/CO₂/H₂. Therefore, a description of the active catalyst as a multiphasic system with ZnO, reduced Zn_{ad}O_x (Zn^{δ+}) and CuZn is in line with our observation, where the reduced and metallic Zn species are predominant under our conditions. In the same NAP-XPS study, ZnO and Zn^{δ+} species were detected on the surface observed even in pure CO₂/H₂, which is in apparent disagreement with our observation of a stable CuZn surface in CO₂/H₂. However, this may be explained by the sequence in which CO was first dosed prior to the NAP-XPS observation⁸ in pure CO₂/H₂, whereby segregation of Zn had already been activated, which could then be oxidized once on the surface.”

2. The general relation to these types of catalyst models to real catalysis is a topic of ongoing discussion in the field. The implications of using simplified surfaces and conditions are well described, but all effects are far from understood. However, in the context of applying new experimental tools like STM and XPS for catalysis studies, the value of conducting a study at significantly challenging conditions closer to catalysis is in general acknowledged in the field. Specifically, the STM method is normally a vacuum technique with far most publications of model catalysts performed under such conditions. STM is in principle not limited by the pressure, but elevated pressure studies with STM are in practice always done at reduced pressure (mbar range typically, with a few examples to 1 bar range) compared with real conditions. There are so far no sophisticated methods available that have studied the surface of a catalyst operating at 20-100 bar, so the comment made by the referee can be used to criticize any paper that tries to use advanced characterization, including already published work. We therefore believe that our first STM studies of the CuZn system carried out at elevated pressures, 6-8 orders of magnitude higher than normal STM, still bears high degree of novelty and relevance.

For instance, could the flat (111) surface not be significant reactive for low dosage of CO₂ at the current pressure and therefore the alloy surface becomes intact and could the low reduction potential of H₂ not be high enough to reduce the partial oxidized Zn Island in the CO₂/CO mixture.

Approaching atmospheric condition, the results may become completely different making the study redundant as being related to methanol synthesis conditions.

Answer: *The main effect we observe in any of the gas mixture is that the CO component induces Zn segregation even when CO₂ is present. It is correct that the more complex starting surface morphologies than a (111) surface could potentially reduce the stability of the Zn species in the Cu surface. However, we do not expect this to be significant, since the Cu(111) and step edges on the surface itself are dynamic in our studies (Figure 2A and 2B), reflecting that a driving force for Zn to be pulled out and become oxidized would be detected.*

Action: *On page 7 we write “The relatively high coordination of Zn atoms in the Cu(111) surface compared to that on stepped surface or a nanoparticle could in principle present a kinetic barrier for Zn abstraction and oxidation, but we exclude this as a significant cause of the apparent stability in the H₂/CO₂ gas since step edges are generated due to the mobility of the Cu surface at the elevated temperature, indicating sufficient mobility.”*

Reviewer #4:

The manuscript deals with the timely topic of the understanding of atomistic processes of methanol synthesis by CuZn based catalysts. The conclusions are drawn from in-situ near ambient pressure STM results and ex situ XPS, as well as UHV STM results. The experimental results are interesting, but there is some additional proof required that the conclusions drawn from the STM image contrast are correct or other possible scenarios need to be discussed.

In addition, the XPS results were obtained under UHV conditions after the in-situ STM experiments, which may have altered the surface and are not directly comparable to the STM observations.

Answer: *The NAP-STM chamber is directly connected to an analysis chamber which contains the XPS analysis system, which means that we avoid transfer through undefined gas conditions or air between the XPS and NAP-STM. XPS is used here to monitor the Cu and Zn chemical state and composition of the surface before and after the experiment, and it was crucial for us to eliminate the presence impurities, which can be a problem for NAP studies in general. The procedure for transferring the sample after STM was to remove the gas in the NAP-STM cell and pump until it could be opened, and the sample was transferred through a gate valve to the XPS without any additional gas exposure. The STM cell is housed inside a vacuum chamber with UHV conditions, which means that this transfer can be carried out efficiently. Although it would have been preferable with in-situ NAP-XPS data (already done in Ref 8), we believe that the post-XPS data concerned with chemical state and composition are precisely reflecting the sample state after the NAP-STM imaging. We note, that even under these ideal UHV transfer conditions we do see some H₂O adsorption from the background during the recording of the XPS data. This is now explained in more detailed in the revised manuscript (see reply to point 4, below).*

Action: *We have clarified the transfer protocol on page 18 (methods).*

Further on, it is debatable that conclusions from an experiment in the mbar regime can be extrapolated to 200 bar pressure. In addition, industrial catalysts are composed of ZnO and Cu nanoparticles. The question arises to which extent studies of a planar model system are relevant for industrial catalysis.

Answer: *This is a generally valid concern for catalyst characterization studies. To avoid repeating the same text, we refer to the answers to reviewer #2 (point 4) and reviewer #3.*

A rigorous comparison with published theory results is lacking, which could further substantiate the hypothesis proposed in this manuscript. One of my main concerns is that the introduced Zn_{ad}-Ox phase lacks experimental evidence.

Answer: *The modelling of O deficient Zn phases has been considered in a number of theoretical studies. We have included the relevant references to theory literature on the topic in relation to our revised discussion of oxidized Zn (see below).*

Action: *On page 10, we have added a sentence: "The presence of partially oxidized Zn on Cu(111), which deviates in structure and composition from bulk ZnO, has indeed been modelled theoretically⁶⁸⁻⁷¹, but mostly in the form of clusters and not the more coherent islands observed here."*

1. On page 4 the STM results after Zn deposition and annealing are described and bright features are assigned to Zn atoms. How can the authors be sure that the model in Fig 6a is correct? Could this also be Zn adatoms or impurities from the deposition? What is the apparent height of these objects? The contrast appears to be much too high to be alloy related, compared to literature results. What are the vacuum conditions during Zn deposition and transfer? Zn is very reactive.

Answer: *The reviewer raises an important point. We agree that STM contrast does not directly reveal the chemical identity of Zn, but it is well known that significant contrast differences arise due to different elements of an alloy. Early work by P. Varga, F. Besenbacher and other groups have shown this for other alloy types, which we also cite. Specifically for Zn in Cu(111), Sano and Nakamura et al has observed it (Ref 35), and we have published a paper (Ref 55) where we investigated the detailed STM contrast in experiment and theoretical simulations for Cu in Zn and Zn in Cu, so we are confident about this assignment.*

The Zn is evaporated from pure Zn using a thermal evaporator kept under UHV conditions in a side chamber to the main chamber. We can safely rule out impurities as the origin of the protrusions, since XPS showed no sign of such, and the Zn amount observed in STM correlates with the observation of Zn in XPS on the surface.

Action: *On page 5 in the figure caption, we now specifically refer to the previous work for the STM contrast: "The STM contrast of Zn species in the Cu(111) surface is reported in Ref. 55"*

2. In a CO₂ / H₂ mixture (from page 5 on), the authors report the formation of Zn dimers, which find very hard to distinguish on the STM images from what the authors assign to single Zn atoms. Would such a phase separation be expected in this low concentration Zn regime from the bulk Cu-Zn phase diagram? How does the Zn 3d spectrum evolve (see Fig 1D), which is sensitive to the Zn distribution in the surface?

Answer: *The Zn3d spectrum in Fig. 1D is a high resolution spectrum recorded on a synchrotron which shows how conversion from a Zn island into Zn alloy leads to more well-resolved peaks. Since the NAP-STM is a stand-alone dedicated experimental station with a Al K α lab X-ray source, we could only record the Zn3d spectrum with limited resolution which generally does not allow us to observe the minute shifts in the 3d spectrum after the NAP-STM (this is one of the reason why we use the Auger ZN LMM signal in Figure 2b). It would be relevant to measure the spectrum as suggested by the reviewer, but even with a synchrotron setup we expect that the dimer formation would not show up as a clear effect, since the broadening is assigned to d-band formation in extended Zn islands. We also refer to the answer to referee 5, point number 4 for an elaboration on the STM contrast associated with the Zn₂ dimers.*

Action: On page 9, we have significantly revised the discussion on XPS and Auger data: “Importantly, the post-analysis of the sample from the Auger LMM peak in Fig. 2C still reflects Zn in the Zn⁰ state, and thus not a Zn²⁺ oxide (see also Supplementary Fig. 8C for the corresponding LMM peaks recorded after pure CO and CO/H₂). The corresponding O1s and C1s XPS data recorded for the sample after NAP-STM in Supplementary Fig. 8A and 8B show no sign of C adsorbates, whereas the O1s indicates some level of O adsorption. The O1s peak is rather broad, and its direct assignment is influenced by O_{ad} species on the Cu from background gas adsorption during the recording of the spectra. Nevertheless, the peak is clearly shifted from the expected 530.3 eV for bulk ZnO on Cu (see also Fig 4B,E), in line with the assignment of Zn_{ad} in a metallic state. We note that the Zn3d peak, which is also sensitive to the formation of Zn_{ad} (as shown in Fig. 1D using synchrotron XPS), was not resolved in high enough detail on the lab-source XPS connected to the NAP-STM.”

3. Under CO₂/H₂/CO mixtures (from page 8 on) the authors state that the images show that a significant amount of Zn has segregated to the surface (line 206). It became not clear so far, that the authors assume that Zn is also subsurface. Can they give some estimate (eg based on the amount deposited and seen by STM), how much Zn is subsurface after the sample preparation? The authors assume that this behavior is related to CO. Here a comparison with literature CO adsorption energies of Cu and Zn might strengthen the argument.

Answer: We thank the reviewer for raising this point, which apparently is the cause of some confusion. We discriminate between Zn in the topmost surface Cu(111) layer and Zn placed onto the surface layer as islands. The way the Zn alloy is made preferentially places Zn only the topmost layer, so subsurface Zn species (i.e. Zn below the visible surface) are not a significant part. The segregation of Zn is from the topmost layer (like in Fig. 2c) into the Zn_{ad} island.

CO-induced surface segregation and aggregation are well-known phenomena, often observed under NAP conditions and have been evidenced on several binary alloys including many Cu-based systems (*J. Phys. Chem. C* **2009**, 113 (33), *Surf. Sci.* **2007**, 601 (23), *Top. Catal.* **2018**, 61 (5–6). *J. Phys. Chem. C* **2019**, 123 (14) on which the segregation typically is seen for the surface constituent binding CO the strongest. We have not found theory papers that specifically model the segregation process we observe, but a study by Mavrikakis and Greeley showed that CO indeed affects the stability of Zn atoms within the surface layers of Cu(111), showing stabilization of Zn (*Journal of Catalysis* 213, 63–72 (2003)).

Action: On page 9 we clarify the type of segregated Zn species as: “...and the images thus remarkably imply that a significant part of the Zn initially present as alloyed Zn atom species in topmost the surface layer has segregated onto the surface in a process driven by CO.”

On page 11 we have further added reference to CO-induced segregation studies and the paper by Greeley, writing: “Greeley et al. also found in theory modelling that CO adsorbates lead to stabilization of Zn atoms in the surface of Cu(111), but the direct extraction of Zn has not been modelled. For CuZn/Cu(111) we therefore speculate that the destabilization of the CuZn surface alloy is driven for example by CO coordinating to Zn in the CuZn-surface, for example forming mobile Zn-carbonyl species.”

4. It is not clear, why the authors assign the new structures observed under CO₂/H₂/CO to a Zn_{ad}-Ox phase. It is rather confusing that this terminology is introduced already from line 230 on, without giving any rational argument. Where is the oxygen coming from? The low pressure CO control experiment does not show the same behavior (page 10 on) put instead at CO pressures in the mbar regime Zn segregation is observed (Fig. S7).

In line 274 the authors state that “While CO₂ is not directly able to induce Zn segregation and oxidize the CuZn alloy...” This is contradicting literature results, in which the oxidation of Cu nanoparticles by CO₂ was reported. The only evidence for oxygen comes from the O1s results in Figure S8 A, which shows a higher signal after CO exposure as compared to after H₂/CO₂/CO exposure. To my opinion, it is therefore very speculative to assign the features observed in STM in CO₂/H₂/CO mixture with a Zn_{ad}-O_x phase. The XPS spectra after low pressure oxidation shown in Fig 4B and C differ significantly from the XPS results shown in Fig. S8 obtained after H₂/CO₂/CO exposure, and there is no direct comparison with the Zn 3d levels available.

Answer. We thank the referee for these comment and will answer them point to point. First, we agree that the way the Zn_{ad}-O_x is mentioned first and discussed should be changed. Therefore, we have significantly revised the discussion on page 9-11 so that we first note our observations and then discuss the possible nature of the Zn_{ad} phase when it is introduced as an explanation for our results, when we perform the Zn oxidation studies that actually prove the existence of such a phase. The purpose of the experiment reported in Figure 4B and C is to show that Zn on Cu(111) can interact with O in two different ways. Oxidation leads to regular (bulk) ZnO monolayer, but milder conditions leads to O adsorption on Zn without any evidence of Zn²⁺. It is important to note, that the main effect we find is that Zn is segregated onto the surface to form the Zn_{ad} islands which only occurs when CO is present. This is supported by the Auger LMM spectra taken after the NAP-STM data, showing no oxidation of the Zn.

The O1s spectra reported in Figure S8a, were recorded after NAP-STM by transferring the sample in vacuum (same chamber). However, as can be observed, the O1s spectrum included O species, but the spectrum is also rather broad, which indicates that several O species. Specifically, we suspect that adsorption of O onto the Cu (e.g. from water in the rest gas during the time it took to record the XPS) plays a role, which complicates the analysis. Nevertheless, the O1s spectrum also extends to the range where the Zn_{ad}-O_x (532.3 eV from Fig 4B), showing that is still a plausible assignment. Combined with the STM data, showing a modification of the Zn, we believe that this is a robust assignment. The most substantial evidence is still the Auger LMM spectra for Zn that shows no Zn²⁺ whatsoever.

Regarding general oxidation of Cu(111) into regular Cu oxide films, it is actually observed that the Cu(111) is not directly prone to oxidation at room temperature in CO₂, even at elevated pressures (J. Am. Chem. Soc. 2016, 138, 26, 8207–8211). This inertness of Cu(111) towards oxidation is further explained in Nature 603, 434-438 (2022). We indeed did a corresponding test experiment for clean Cu(111) in 10 mbar CO₂, and the terraces stay metallic, except for some oxidation near steps. This may be because step edges are active in CO₂ activation, but the amount of surface O species from the step in our case is not dominating the effect seen here when Zn is added.

Action: The discussion on page 9-11 is significantly changed, so that it follows the suggestion from the reviewer. We have added a further discussion of the O1s spectra (Figure S8), and the link to the Zn_{ad}-O_x phase is now only discussed in the following section where Figure 4 is explained.

Reviewer #5 (Remarks to the Author):

1. Fig. 1D, black spectrum: The intensity ratio of the Zn 3d(5/2) : 3d(3/2) peaks seem to be lower than the expected 3:2. Please explain.

Answer: There are two convoluted effects that affect the Zn3d doublet ratios. Firstly, Zn is a relatively light element and Zn3d is a shallow core level. The spin-orbit splitting is 0.35eV, meaning the peaks are not fully resolved as individual peaks even with the synchrotron setup. We have done the peak fitting, and it indeed gives a ratio which deviates by 15% from the ideal (2:3). Final state effects such as photoelectron diffraction could be a reason for this (see Himpsel et al, Phys Rev B 22, 10 (1980)). Secondly the samples are made by alloying Zn from monolayer islands into the surface where ideally, they are distributed as isolated Zn species, but remnant minor Zn species on the surface or Zn placed on step edges etc. could lead to a small chemical shift of the involved Zn species and therefore some broadening towards the high binding energy side which is difficult to fit precisely. The shift in the peak shape is however evident, and we do not believe that the possible deviation in the ratio has implications for our interpretation of Zn being present mainly as islands or Zn atom in the surface .

2. The tunneling voltage and current data are missing in Fig. 1C.

Answer: Thanks. They are added to the figure caption.

3. Fig. 2A and B: What is the driving force of the morphological change? The presence of the H₂/CO₂ gas phase may explain enhanced mobility, but it cannot constitute the driving force, a thermodynamic property, in contrast to what the authors suggest. The same holds for the discussion on page 10, top paragraph.

Answer: We thank the reviewer for this interesting question. We observe indeed that the Cu surface restructures when exposed to gases at elevated temperatures. We attribute the underlying reason to gas-induced diffusion, which is much more pronounced at elevated pressure than in regular UHV studies. Yet, Cu(111) even as clean surface is dynamic, seen by the fluctuation of step edges observed over sometime. This is well understood and is due to emission of Cu ad-atoms from kinks. The STM work by Harrington (Ref 59) demonstrates how adsorbates can accelerate this process many times and modify the surface morphology considerably even by UHV exposure to gases.

The reviewer is right that the free energy of the (111) surface should not be affected. In our case, the reason why the surface morphology changes from relatively flat could be due to entropy in involved, i.e. in line with the roughening transition that occurs when Cu surfaces are over-annealed (again affecting the Cu mobility). Alternatively, the preferential adsorption of some adsorbates on edge sites compared with terraces, which over time promotes the formation of more steps on the dynamic Cu surface.

Action: On page 6 we write: "The observed surface roughening is likely driven by surface mobility of emitted Cu adatoms, and we speculate that either entropy effects or preferential stabilization of step edges either from CO₂ or HCOO could provide a driving force."

4. Fig. 2C: It appears that most of the Zn dimers point into the same direction, which may indicate a double tip, a frequent artifact in STM data. Whether such an artifact is present could easily be decided by determining the distribution of the orientational angles of the dimers. True dimers should display random orientations.

Answer: The reviewer raises an important point. It is quite rare to report atom-resolved STM images recorded at mbar pressure, and the challenging imaging conditions means that STM images under such conditions may reflect asymmetric tip geometries. In the case of Figure 2C, the tip seems to be slightly asymmetric which means that some of the dimers stand out with high contrast than other. From our detailed UHV-STM studies of Zn atoms in Cu(111) (see e.g. Ref 55), we found that the contrast of Zn even under such conditions was sometimes complex. The important point in Fig. 2C

and the images from that series (see also Supplementary Figure 5) is however, that we can identify both Zn monomers and dimers of all 3 symmetry orientations, which would not be possible if the dimers are an effect of a double tip. To make this clearer the example figure below shows Zn₂ dimers of all 3 high symmetry directions on the Cu(111), ruling out a double tip effect.

Action: *In figure S5, we have added “We note that the Zn site STM contrast showed some variation dependent on the tip state even in UHV. The STM contrast in (C,D) shows sign of a complex imaging mode, but importantly Zn₂ dimers can be found in all three symmetry orientations of the Cu(111) surface, ruling out an effect of a double tip.”*

5. Page 6 and Fig. S4. A statement is made that CO₂ alone does not lead to dealloying. What does H₂ alone do?

Answer: *We did not carry out an experiment with pure H₂ dosing, since we expected that the Zn would respond primarily to the CO₂ (oxidation) or CO component of the gas. H₂ is a reducing gas that may prevent oxidation, so the pure CO₂ experiment was performed to check if an elevated chemical potential of H₂ in the mixed gas kept the CO₂ from oxidizing the surface, but this seems not to be the case.*

Action: *On page 7: “Moreover, since the CuZn is stable in CO₂ without H₂ gas, it also seems that fast reduction of surface O species by the H₂ is not a dominant effect.”*

6. Fig. 3A. The coverage of Zn islands was 0.08 +/-0.03 ML. It is not a trivial matter to obtain good statistics of the coverages of structures that are similarly large as the diameters of the STM images. Please explain the method. How many images from different areas have been analyzed? Have automatic tools been used to determine the areas?

Answer. *We thank the reviewer for both suggestions. The Zn islands are relatively easy to discriminate from their height relative to Cu(111) (see also Ref 55 and Figure S1). Also, since the Zn assembles in rather large islands like in Fig. 3A, we believe it is not required to use automatic detection tools to get an estimate of their coverage. The number is an estimate of the Zn segregated onto the surface, which we can compare with the amount of Zn deposited. Here we note that the number is lower than the 0.15 ML deposited on the surface, and this might indeed be due to*

statistics. We have now noted this in the paper, but also note that the amount of Zn is consistent with some remnant Zn in the surface and the precision involved in estimating coverages from STM.

Action: On page 9 we write “The lower coverage of Zn detected in NAP-STM is attributed to remnant Zn in the surface and also that the Zn assembles into rather large islands which affects the precision in the estimate of the average coverage from STM images alone”

From the area covered by Zn islands and the coverage of O atoms obtained from the O 1s peak in XPS one can evaluate the local coverage of O atoms on the islands. This might be an important parameter that discriminates the ZnO_x islands from stoichiometric ZnO.

We agree that the O area in the O1s spectra should reflect the O content, and we have done the estimation now. Before explaining this, it is important to note that the O1s spectrum in Figure S8 is broad, clearly reflecting more than one O species. We have indicated in the spectrum where O in Zn_{ad}O_x and ZnO should be located (as in Figure 4). Our new analysis also concludes that some of the O results from background adsorption onto Cu, e.g. from residual H₂O. This would be a peak located at 530.9 (see Eren et al, J. Phys. Chem. C 120, 8227–8231 (2016)). The spectra are recorded with a lab-source which means that several hours are needed to first transfer and then to record a properly measured spectrum. There is some variation from experiment to experiment, but in all cases the O amounts are low.

To compare, an estimation of the O peak area on the CuZn(111) surfaces for was performed by a separate reference experiment where Cu-oxide was grown on the Cu(110) facet ((2x1)-Cu(111)). On the Cu(110) facet the oxide growth self-terminates after a 0.5 ML is formed, hence the very well-defined reference experiment can be used to estimate the O coverage on the Zn-Cu(111) surface by simply evaluating the ratio. The quantification shows that the O to Zn ratio is from 30 to 90% relative to Zn (0.15ML), i.e. always less than 100%, even when the O on the Cu is included. To single out the O in the Zn-phase, we next tried to fit the data according to possible species, focusing on the part of the spectrum that should reflect Zn_{ad}O_x. The figure below shows this peak (blue) and the number in all cases show that the O level associated with Zn is in the range from 0.03-0.05ML O (i.e. 20-30% O content relative to Zn). This is enough to say that any Zn phase is reduced relative to stoichiometric.

Action: Overall, based on the input from this reviewer and reviewer #4 we have thoroughly revised the discussion on page 11-13 related with the Zn_{ad} phase and the observations that suggest the presence of O adsorbates. We have also included detailed information related to the O1s spectra (Figure S8), where we discuss the difficulties in assigning the O1s peak component due to likely O adsorbates on the Cu. We have chosen not to include the quantification, explained above, in the revised manuscript, and we hope the reviewer will agree to this choice. The main reason is the uncertainties in fitting the spectrum and the role of the O background adsorption. Moreover, the Auger LMM is rather clear in assigning the Zn oxidation state to metallic Zn and the STM data also points to adsorbates on the Zn islands. In this way the results related to Figure 4 is used to argue that the Zn islands are partially oxygen covered, but still reflect by Zn⁰ in the spectroscopy.

7. The caption of Fig. 3D is missing; 3E is not marked.

Answer. We thank the reviewer for noting this. Figure 3D are the inserts from Figure 3B, but we have removed the D and now note that the inserts below correspond to the squares marked in the image.

8. Have any experiments been performed to analyze whether the dealloying was reversible, i.e., did the islands disappear when the gas was changed from H₂/CO₂/CO to H₂/CO₂?

Answer. We have not done this experiment, but it raises an important point. The temperature to go from Zn islands to alloy in UHV was studied by us in Ref 55, and temperatures of 480K are needed to fully convert Zn islands into CuZn. From this we would not expect the alloy to form in the suggested experiment. However, the work by Amann (Ref 8), who uses NAP-XPS to study a similar CuZn model

surface has investigate the switch from CO rich to CO₂ rich gas composition, and they do not detect switch back to a CuZn alloy (Figure 2 in Ref 55).

9. On page 10, top paragraph, it is argued that the Zn islands on the alloy rim around the islands are active. It is not clear to me on which observation this is based. Couldn't the islands themselves be active?

Answer. We thank the reviewer for this suggestion. Our aim with this work was to clarify the surface structures and to which extent the two main models for the active phase based on CuZn or ZnOx/Cu are stable surfaces in the gases used for methanol synthesis. From our methods we cannot determine the actual active sites and we do not want to claim this. The discussion is mainly a reflection of our observation noting that Zn species are moved in and out of the CuZn alloy in this region, which overall reduces the amount of the surface which is covered by CuZn.

Action: We have removed the discussion in this part of the manuscript, since it is better discussed in the discussion section, which we have also revised and shortened in response to reviewer's comments.

10. Altogether, I find the language and terminology somewhat inexact. On page 10 a statement is made that "... an equilibrium is established [...] in the vicinity of the island edges", and a similar statement is made in the abstract. The term equilibrium suggests a permanent exchange of Zn atoms between the edges and the islands, but this is not was has been observed.

Answer. We thank the reviewer for noting this and have revised the section on page 10-11 to explain the situation better. The term equilibrium here refers to the balance between two processes which are active in the CO containing gas: (1) CO-induced segregation of Zn from the alloy onto the surface and (2) the reverse process where Zn located at the perimeter is exchanged into the Cu(111) surface (driven by temperature). Process 2 was investigated in Ref 55, and is responsible for the formation of the CuZn zone near the Zn island shown for UHV studies in Figure 1D. We acknowledge that the explanation for process 2 was not sufficiently clear (see also reply to reviewer #2, point 5) and we have revised the paragraph that describes this, avoiding the term equilibrium.

Action: On page 11: "Our interpretation of this is that a dynamic situation is present in NAP-STM syngas conditions where two processes are active and affecting the Zn. The first is the CO-induced segregation of Zn from the CuZn surface which supplies Zn to the Zn_{ad} islands, but once the Zn_{ad} phase is formed, a second process is established where Zn is supplied back to the Cu surface in the vicinity of the island edges. CO-induced surface segregation is a well-known phenomenon, evidenced on several binary alloys^{72,73,74} including many Cu-based systems^{75,76,77}. Greeley et al. also found in theory modelling that CO adsorbates lead to stabilization of Zn atoms in the surface of Cu(111)⁷⁸, but the direct extraction of Zn was not modelled. For CuZn/Cu(111) we therefore speculate that the destabilization of the CuZn surface alloy is driven for example by CO coordinating to Zn in the CuZn-surface, for example forming mobile Zn-carbonyl species.

11. The term "AP-STM", used throughout the ms, is not correct. AP (ambient pressure) means 1000 mbar, not the more than two orders of magnitude lower pressures applied in the experiments. That the term has been used in previous work at similarly low pressures is not a sufficient argument.

Answer. We agree with the reviewer. We did not intend to overemphasize the applied pressure range in our study. The literature uses AP-STM and NAP-STM as terms interchangeably (even high-pressure STM), and we are also aware about the variation in the general use of these terms. We used AP-STM

for consistency with relevant literature on Cu, but near-ambient pressure (NAP) is indeed a more appropriate term here.

Action. *AP-STM has been changed to NAP-STM throughout, and we explain in page 2 in the introduction that other terms are used for the same thing.*

12. The claim made in the abstract that the work "... establishes a complex role of CO as an activator of catalytic activity for methanol formation and as an inhibitor ..." is not really based on the data because no activity has been measured in this work that could justify such a conclusion.

Answer. *We have revised this. The main finding is that CO gas is the main factor that affects how Zn is distributed on the Cu surface.*

Action: *In the abstract: "The results show that the Zn distribution on Cu(111) is strongly sensitive to the CO component in the gas at mbar pressures, and thus proposes a significant role of CO affecting the distribution of Zn in a multiphasic ZnO/CuZn/Cu catalysts."*

REVIEWER COMMENTS

Reviewer #2 (Remarks to the Author):

My doubts have basically been resolved. I think the findings of this research work are worth publishing in Nature Communications.

Reviewer #3 (Remarks to the Author):

I find that the authors have addressed my comments in a satisfactory way. They have toned down some their claims and discuss more openly that they are far away from the methanol synthesis conditions. I find study valuable as a model study of th Cu-Zn system with gas interactions. I now recommend publications of the study.

Reviewer #4 (Remarks to the Author):

I have reviewed the reply to my comments and the manuscript and I am happy with the changes performed. I can now recommend publication.

Reviewer #5 (Remarks to the Author):

I shortly comment on the reports by the other reviewers. Their most fundamental critique aims at the fact that the investigation was performed in the mbar range, 4 orders of magnitude below the pressures applied in the actual methanol synthesis, and that the state of the surface observed by STM could not be correlated with the catalytic activity. This is certainly true. On the other hand, one has to acknowledge that it is still a major challenge obtaining information on the surface of a catalyst under actual reaction conditions. In my view, the study of Jensen et al. at least makes a significant step in this direction for the methanol synthesis and is therefore meaningful for catalysis. My general assessment was therefore positive.

However, I had several technical objections. Most of these have adequately been resolved in the revised version of the ms or by the rebuttal by the authors. However, two issues have not been clarified:
/1/ I had doubts about the claim that Figure 2C mainly shows Zn dimers. Most of the dimers point into one of three possible directions which indicates an artifact from a double tip. In their response, the authors say that some of the dimers also point into other directions. However, what I was asking for was a statistical analysis of all dimers in the entire image, not of just a few of them. The authors should count in the full image how many dimers point into the three possible directions. If this distribution is not uniform within statistical errors, this would be clear evidence for a tip artifact. In this case the aspect of the Zn dimers would have to be removed from the ms.

/2/ I had asked about the data basis on which the value for the coverage of Zn islands is based. Using Figure 3A, the authors determined a value of 0.08 ± 0.03 ML but do not provide the data basis. In their response, the authors say that it is "... relatively easy to discriminate [Zn islands] from their height ...". This is clear but not what my question was about. What I had asked was how many STM images have been analyzed to evaluate the coverage. When a topographic feature in a microscopic image is similarly large as the diameter of the image, which is the case in Figure 3A, then the relative area covered by this feature is not a statistically relevant number. In this case, one has to analyze more than one image, say 10 or 20, from diverse areas, to obtain a meaningful number. When these two issues have been clarified I could recommend publication of this ms.

Reply to Reviewer #2

My doubts have basically been resolved. I think the findings of this research work are worth publishing in Nature Communications.

Answer: Thank you

Reviewer #3

I find that the authors have addressed my comments in a satisfactory way. They have toned down some their claims and discuss more openly that they are far away from the methanol synthesis conditions. I find study valuable as a model study of th Cu-Zn system with gas interactions. I now recommend publications of the study.

Reply: Thank you

Reviewer #4

I have reviewed the reply to my comments and the manuscript and I am happy with the changes performed. I can now recommend publication.

Reply: Thank you

Reviewer #5

I shortly comment on the reports by the other reviewers. Their most fundamental critique aims at the fact that the investigation was performed in the mbar range, 4 orders of magnitude below the pressures applied in the actual methanol synthesis, and that the state of the surface observed by STM could not be correlated with the catalytic activity. This is certainly true. On the other hand, one has to acknowledge that it is still a major challenge obtaining information on the surface of a catalyst under actual reaction conditions. In my view, the study of Jensen et al. at least makes a significant step in this direction for the methanol synthesis and is therefore meaningful for catalysis. My general assessment was therefore positive.

However, I had several technical objections. Most of these have adequately been resolved in the revised version of the ms or by the rebuttal by the authors. However, two issues have not been clarified:

1. I had doubts about the claim that Figure 2C mainly shows Zn dimers. Most of the dimers point into one of three possible directions which indicates an artifact from a double tip. In their response, the authors say that some of the dimers also point into other directions. However, what I was asking for was a statistical analysis of all dimers in the entire image, not of just a few of them. The authors should count in the full image how many dimers point into the three possible directions. If this distribution is not uniform within statistical errors, this would be clear evidence for a tip artifact. In this case the aspect of the Zn dimers would have to be removed from the ms.

Reply: The reviewer questions the assignment of Zn dimers in Figure 2C. We disagree with the reviewer that such detailed statistics are needed to rule out a double tip effect here, since dimers are observed in all directions, as we argued in our first reply to this concern. However, at the same time,

it is clear that a direct inspection of the image by the readers could lead to the impression of a double tip, which without a proper explanation, would be of concern for the discussion. For atom-resolved images, as in this case, the observed brightness of dimers in certain directions could be an effect of an asymmetric tip that enhances the contrast mainly in some directions. To avoid a lengthy discussion of this technical topic in the paper, we feel that it is better to remove the discussion of dimers related to figure 2C (and supplementary figure 5). We will publish them instead in another context. This change to the paper can be done without changing any of the main conclusions of the paper.

Action: The paragraph on page 7 concerning the analysis of Zn dimers has been removed together with Supplementary figure 2.

2. I had asked about the data basis on which the value for the coverage of Zn islands is based. Using Figure 3A, the authors determined a value of 0.08 ± 0.03 ML but do not provide the data basis. In their response, the authors say that it is "... relatively easy to discriminate [Zn islands] from their height ...". This is clear but not what my question was about. What I had asked was how many STM images have been analyzed to evaluate the coverage. When a topographic feature in a microscopic image is similarly large as the diameter of the image, which is the case in Figure 3A, then the relative area covered by this feature is not a statistically relevant number. In this case, one has to analyze more than one image, say 10 or 20, from diverse areas, to obtain a meaningful number.

Reply: The referee asks for a more rigorous statistical analysis of the determined value of 0.08 ± 0.03 ML Zn coverage. The referee is right that a precise estimate of this value requires a deeper statistical analysis, which in our case is complicated by the limited number of images that can be recorded for these demanding experiments, and the strong tendency for Zn to grow as few, but large islands. Nevertheless, we consider that the observed amount in our available images is always less than, and not higher, than the original 0.15 ML. The discussion contributes to the understanding that they are Zn species segregating out of the bulk. We did not intend to report the value with a higher significance than that we have. In the revision we have therefore revised the discussion to reflect this point.

Action: On page 9 we have revised the paragraph which is now written as: "The Zn island coverage (~ 0.08 ML in Fig. 3A) was always lower, and never higher, than the initially deposited amount of Zn (~ 0.15 ML), which is consistent with a process where Zn segregates out and nucleates and grows into the observed islands. We note that the Zn assembles into rather large islands which affects the precision of the estimate of the average Zn coverage from available NAP-STM images recorded at these conditions, explaining why not all Zn contained in the initial alloy is accounted for."

REVIEWERS' COMMENTS

Reviewer #5 (Remarks to the Author):

The issue of the Zn dimers has been removed from the text which resolves the first of my two remaining points of critique. However, it is still present in Fig. 2C in the labeling, and it is still contained in the figure caption: "The white arrows highlight cases for monomeric Zn₁ and dimeric Zn₂ species." Consistent changes are required. The second issue, the value of the coverage of the Zn islands has been properly corrected.

Reviewer #5

The issue of the Zn dimers has been removed from the text which resolves the first of my two remaining points of critique. However, it is still present in Fig. 2C in the labeling, and it is still contained in the figure caption: "The white arrows highlight cases for monomeric Zn1 and dimeric Zn2 species." Consistent changes are required. The second issue, the value of the coverage of the Zn islands has been properly corrected.

Reply: *We have removed the marked arrows in the figure and the associated text to the discussion of Zn dimers.*